# Pharmacogenomics of poor drug metabolism in greyhounds: Canine P450 oxidoreductase genetic variation, breed heterogeneity, and functional characterization

**Stephanie E. Martinez** [1¤], **Amit V. Pandey**[2], **Tania E. Perez Jimenez**[1], **Zhaohui Zhu**[1], **Michael H. Court** [1]*

**1** Pharmacogenomics Laboratory, Program in Individualized Medicine (PrIMe), Department of Veterinary Clinical Sciences, College of Veterinary Medicine, Washington State University, Pullman, Washington, United States of America, **2** Pediatric Endocrinology, Diabetology, and Metabolism, Department of Biomedical Research, University Children's Hospital Bern, Switzerland and Translational Hormone Research Program, University of Bern, Bern, Switzerland

¤ Current address: Department of Anatomy and Physiology, College of Veterinary Medicine, Kansas State University, Manhattan, Kansas, United States of America
* michael.court@wsu.edu

**Data Availability Statement:** Computer model data used for the POR dynamic structural modelling are

## Abstract

Greyhounds metabolize cytochrome P450 (CYP) 2B11 substrates more slowly than other dog breeds. However, *CYP2B11* gene variants associated with decreased *CYP2B11* expression do not fully explain reduced CYP2B11 activity in this breed. P450 oxidoreductase (POR) is an essential redox partner for all CYPs. POR protein variants can enhance or repress CYP enzyme function in a CYP isoform and substrate dependent manner. The study objectives were to identify POR protein variants in greyhounds and determine their effect on coexpressed CYP2B11 and CYP2D15 enzyme function. Gene sequencing identified two missense variants (Glu315Gln and Asp570Glu) forming four alleles, POR-H1 (reference), POR-H2 (570Glu), POR-H3 (315Gln, 570Glu) and POR-H4 (315Gln). Out of 68 dog breeds surveyed, POR-H2 was widely distributed across multiple breeds, while POR-H3 was largely restricted to greyhounds and Scottish deerhounds (35% allele frequencies), and POR-H4 was rare. Three-dimensional protein structure modelling indicated significant effects of Glu315Gln (but not Asp570Glu) on protein flexibility through loss of a salt bridge between Glu315 and Arg519. Recombinant POR-H1 (reference) and each POR variant (H2-H4) were expressed alone or with CYP2B11 or CYP2D15 in insect cells. No substantial effects on POR protein expression or enzyme activity (cytochrome c reduction) were observed for any POR variant (versus POR-H1) when expressed alone or with CYP2B11 or CYP2D15. Furthermore, there were no effects on CYP2B11 or CYP2D15 protein expression, or on CYP2D15 enzyme kinetics by any POR variant (versus POR-H1). However, $V_{max}$ values for 7-benzyloxyresorufin, propofol and bupropion oxidation by CYP2B11 were significantly reduced by coexpression with POR-H3 (by 34–37%) and POR-H4 (by 65–72%) compared with POR-H1. $K_m$ values were unaffected. Our results indicate that the Glu315Gln mutation (common to POR-H3 and POR-H4) reduces CYP2B11 enzyme

available here https://doi.org/10.48620/379. All other underlying data, including blot images, used to generate the graphs and tables shown in the manuscript are provided in the Supporting Information (S1 File).

**Funding:** S.E.M. and M.H.C. were funded by American Kennel Club Canine Health Foundation (Grants 2242, 2529; https://www.akcchf.org/). M. H.C. received funding from the William R. Jones Endowment at Washington State University College of Veterinary Medicine (https://vetmed.wsu.edu/). A.V.P. was funded by grants from the Swiss National Science Foundation (310030M_204518; https://www.snf.ch/en), Novartis Foundation for Biomedical Research (https://www. novartisfoundation.org/), and Bern University Research Foundation (https://www.unibe.ch/ index_eng.html). The funders had no role in study design, data collection and analysis, decision to publish, or preparation of the manuscript.

**Competing interests:** The authors have declared that no competing interests exist.

function without affecting at least one other major canine hepatic P450 (CYP2D15). Additional *in vivo* studies are warranted to confirm these findings.

## Introduction

Accumulated evidence over the last 30 years has shown that metabolism rates for some drugs are markedly slower in greyhounds than in other dog breeds [1–8]. Initial pharmacokinetic and pharmacodynamic studies demonstrated that greyhounds recovered three to four times slower from lipophilic thiobarbiturates (thiopental and thiamylal) and possessed significantly higher plasma drug concentrations than mixed-breed dogs receiving the same doses [9, 10]. However, when greyhounds and mixed-breed dogs were administered oxybarbiturate anesthetics (pentobarbital and methohexital) with similar lipophilicities as thiobarbiturates but with slightly different chemical structures, similar plasma pharmacokinetic parameters and recovery times between the two groups of dogs were observed [9]. A subsequent study showed that when greyhounds were treated with phenobarbital, a cytochrome P450 (CYP) inducer, thiopental clearance increased, and recovery times were reduced compared to untreated greyhounds [11]. While propofol has largely replaced thiobarbiturates for routine induction of anesthesia in dogs, the clearance of propofol and recovery times have also been shown to be slower in greyhounds than in other dog breeds [12, 13]. Furthermore, when greyhounds were administered chloramphenicol, a CYP inhibitor, dogs had reduced propofol clearance and prolonged recovery times compared to dogs administered propofol without chloramphenicol [14]. These studies implicate CYP-mediated elimination of these drugs and suggest that greyhound could be deficient in one or more CYP isoforms.

Previous studies in our laboratory showed that propofol is primarily metabolized by CYP2B11 and that liver microsomes from greyhounds metabolize CYP2B11 substrates, including propofol and bupropion, slower, and possess less hepatic CYP2B11 than liver microsomes from other breeds of dogs [2,4]. However, greyhound livers were not deficient in abundance or activity of any other major drug metabolizing CYP. Sequencing of the CYP2B11 gene in greyhounds discovered two variant *CYP2B11* haplotypes (*CYP2B11*-H2 and -H3) differentiated by mutations in the 3'-untranslated region of the gene [2]. No other variants were discovered. *In vitro* experiments using a 3'-untranslated region reporter system indicated that these *CYP2B11* mutations decrease enzyme expression through reduced translational efficiency. The *CYP2B11*-H2 and -H3 variant haplotypes showed reduced reporter gene expression by 40% and 70%, respectively, compared to H1.

Genotyping over 2,000 dogs representing 64 different dog breeds and mixed-breed dogs primarily from the United States, showed that the *CYP2B11*-H2 allele was distributed across many breeds, while the *CYP2B11*-H3 allele was primarily localized to the sighthound breeds. Sighthounds (that include greyhounds) are a grouping of dog breeds that share many phenotypic characteristics in part because of a shared genetic heritage [15]. Sighthounds are so-called because they were bred to hunt prey by sight rather than by scent. Greyhounds were found to have the highest *CYP2B11*-H3 allele frequency among the breeds examined (28%). However, when greyhounds were separated into subgroups of dogs registered with the National Greyhound Association (NGA) bred for racing or dogs registered with the American Kennel Club (AKC) bred for conformation, the *CYP2B11*-H3 allele frequencies were 17% and 59%, respectively. Additionally, while NGA-registered greyhounds have a *CYP2B11*-H2 allele frequency of 26%, the allele was not found in AKC-registered greyhounds.

There are far more NGA-registered than AKC-registered greyhounds in the United States due to the greyhound racing industry [16, 17]. With an approximate *CYP2B11*-H3/H3

diplotype frequency of 35%, reduced CYP2B11-mediated drug metabolism in AKC-registered greyhounds may be largely explained by *CYP2B11*-H3. However, the 3% *CYP2B11*-H3/H3 and 7% *CYP2B11*-H2/H2 diplotype frequencies in NGA-registered greyhounds cannot fully explain the slow drug metabolism phenotype observed in these dogs. Consequently, other mutations in CYP-interacting genes that have the potential to affect CYP2B11 function, without affecting the function of other drug-metabolizing CYP isoforms, should be considered.

P450 oxidoreductase (POR) is an essential microsomal diflavoprotein that donates electrons to all microsomal CYP enzymes for their catalytic activities, as well as cytochrome $b_5$, heme oxygenase, squalene monooxygenase and bioreductive prodrugs [18]. Complete *POR* gene knockout in mice is embryonic lethal, whereas liver-specific *POR* knockout mice appear phenotypically normal but show a marked decrease in drug metabolism [18]. *POR* is highly polymorphic in humans, with over 200 different mutations and polymorphisms identified [18, 19]. In humans, the most severe *POR* mutations cause a loss of cofactor binding sites in the enzyme, leading to near inactivation of POR and P450 oxidoreductase deficiency disorder, a form of congenital adrenal hyperplasia [18, 20–22]. Of the *POR* polymorphisms investigated that do not result in the loss of cofactor binding, variable disruptions to POR and partner enzyme activities have been observed, including both enhancement and repression for the same enzyme variant [23]. Perhaps the best example is *POR*28*, a conservative missense variant (A503V) with a human population allele frequency between 19 and 40%. *In vitro* functional analyses showed decreased enzyme activities for POR itself, and coexpressed CYP2D6, CYP1A2, and CYP17A1, while CYP2C19 and CYP2C9 activities were increased, and heme oxygenase activities were unchanged, compared with wild-type (reference) *POR* [19, 23]. Interestingly, the effects of the A503V substitution on CYP3A4 catalytic activity appear to be substrate-dependent with both enhancement and repression [19, 23]. Given that the effects of *POR* mutations can be influenced by their redox partners in humans, it is possible that one or more missense mutations in the dog *POR* gene could affect CYP2B11-mediated metabolism without affecting metabolism mediated by other canine CYP isoforms.

The primary objectives of this study were to identify *POR* gene mutations that may contribute to reduced CYP2B11-mediated metabolism in greyhounds and determine the functional effects of *POR* mutations on POR enzyme function and on the function of coexpressed CYP2B11 and CYP2D15 [24, 25]. CYP2D15 was evaluated since a previous study showed that this CYP is not deficient in greyhounds compared with other breeds [26]. The distribution of the identified *POR* mutations across dog breeds was also explored. We hypothesized that *POR* mutations would be more prevalent in greyhounds and closely related sighthound breeds than in non-sighthound breeds, and that these *POR* mutations would decrease the function of coexpressed CYP2B11 but not CYP2D15.

## Materials and methods

### Animal ethics statement

The collection and use of liver tissues employed in this study were considered exempt from review by the Institutional Animal Care and Use Committee at Washington State University since all tissues collected would have normally been discarded. The collection, storage, and use of the DNA samples employed in this study were approved by the Institutional Animal Care and Use Committee at Washington State University (protocols #04194 and #04539) and were conducted in accordance with the relevant guidelines and regulations. Informed owner consent was obtained for all dogs before DNA collection.

## Chemical and reagents

Tween®-20, glycerol, HPLC-grade acetonitrile, and methanol were purchased from Thermo Fisher Scientific (Waltham, MA, USA). Formic acid, hydrochloric acid, ethanol, glacial acetic acid, sodium hydroxide, sodium chloride, Tris base, magnesium chloride, potassium phosphate monobasic, potassium phosphate dibasic, and EDTA were purchased from J.T. Baker (Center Valley, PA, USA). Cytochrome c isolated from porcine heart was purchased from Gold Biotechnology, Inc. (St. Louis, MO, USA). 7-Benzyloxyresorufin (BROD), resorufin, NADP$^+$, isocitrate dehydrogenase, $_{DL}$-isocitrate, dextromethorphan, dextrorphan, thymol, sodium cyanide, and Coomassie® Brilliant blue R-250 were purchased from Sigma-Aldrich (St. Louis, MO, USA). Tramadol hydrochloride, O-desmethyltramadol hydrocholoride, O-desmethyltramadol-d6, and 4-hydroxypropofol were obtained from Toronto Research Chemical Inc. (Toronto, ON, Canada). Bupropion hydrochloride, hydroxybupropion, and GW340416A, a chemical analog of bupropion, were kindly provided by GlaxoSmithKline (Research Triangle Park, NC). Propofol was provided by Zeneca Pharmaceuticals (Wilmington, DE, USA). Hemin from a porcine source was purchased from BeanTown Chemical, Inc. (Hudson, NH, USA). Beta-NADPH tetrasodium salt was purchased from Alfa Aesar (Haverhill, MA, USA). Ultra-pure water was obtained using a Milli-Q® Q-POD Millipore system (EMD Millipore, Burlington, MA, USA).

## Dog breed DNA sampling

DNA samples (n = 2,286) from privately owned dogs were retrieved from the Washington State University Veterinary Teaching Hospital Patient DNA Bank and Pharmacogenomics Laboratory Sighthound DNA Bank. DNA was extracted from buccal swab samples obtained from hospital staff or by the owner of the dog. Most of the hospital patient population samples were from dogs living in the Pacific Northwest of the United States whereas the sighthound samples were obtained primarily by mail from dogs living throughout the United States. A dog's breed was identified by the owner for samples from the Hospital Bank whereas the breed was identified by the owner along with the accompanying breed registration identification for samples from the Sighthound Bank. Dogs whose breeds were designated as 'mix,' 'mixed,' 'cross,' 'mutt,' 'mongrel,' or similar by the owner were considered as a single group of 'mixed-breed' dogs for this study (n = 148). The 68 different dog breeds sampled included 21 sighthound breeds and 47 other breeds. The designation of a breed as belonging to the 'sighthound' group was based on the AKC's breed inclusion for sighthounds[26]. Samples from greyhound dogs (n = 258) were divided into two groups based on whether they were identified by their owners as dogs bred for racing and registered with the NGA (n = 196) or dogs bred for other purposes and registered with the AKC (n = 62). All breed groups included samples from at least 10 dogs.

## *POR* sequencing and genotyping

*POR* exons 1–16 were sequenced by Sanger sequencing to identify genetic polymorphisms in genomic PCR product from DNA obtained from 13 greyhounds (five from liver samples and eight from buccal swab samples) and 5 beagles (all from liver samples). Primers used for PCR and sequencing, gene region amplified, and product size are given in S1 Table. Custom allele discrimination assays (Applied Biosystems TaqMan SNP Genotyping Assay, Thermo Fisher Scientific) were used to genotype DNA samples from all 2,286 dogs for the *POR* haplotype associated single nucleotide polymorphisms (SNPs): c.943 G/C and c.1710 C/G. Primer and reporter sequences are given in S2 Table. Assays were performed using a real-time PCR instrument (CFX96 Touch, Bio-Rad, Hercules, CA, USA). Genotype frequencies for each SNP were

calculated by dividing the number of variant alleles by the total number of alleles for each breed. Diplotype (haplotype pair) designations for each dog were inferred from the c.943 G/C and c.1710 C/G genotype data. Possible haplotypes included POR-H1 (reference), POR-H2 (c.1710G), POR-H3 (c.943C and c.1710G), and POR-H4 (c.943C). For the purposes of calculating haplotype frequencies, dogs that were heterozygous for both SNPs (c.943 GC and c.1710 CG) were assumed to have the POR-H1/H3 diplotype rather than POR-H2/H4, given the low frequency of the unambiguous POR-H1/H4 and *POR*-H4/H4 diplotypes compared to the POR-H1/H2 and POR-H2/H2 diplotypes in the population studied.

### *In silico* analyses

Initial analysis included multiple sequence alignment of canine POR with homologs from other species was also used to determine sequence homology and conservation at amino acid substitution sites [27]. PolyPhen-2 (Polymorphism Phenotyping v2) was then used to qualitatively predict the potential impact of the amino acid substitutions on the structure and function of the POR protein [28]. Three-dimensional structural analysis was then performed using five different x-ray crystal structures of POR (PDB: 3QE2 human, 3QFC human, 4YAL rat, 5URD rat and 1JA1 rat) to generate models of canine POR which were then combined to generate a hybrid model retaining the best scoring parts of individual structures. The selection of templates was based on BLAST alignment scores the WHAT_CHECK [29] quality score in the PDBFinder2 database [30] and the target coverage. For alignment correction and loop modeling, a secondary structure prediction for the target sequence was obtained by running PSI-BLAST to create a target sequence profile and feeding it to the PSI-Pred secondary structure prediction algorithm [31]. The stability of mutant proteins was analyzed using DUET [32] SDM [33] and DynaMut2 [34] and by FoldX analysis [35] of WT and mutant proteins running as a python script under Yasara [36].

### Recombinant POR and CYP expression

cDNA sequences for canine *POR*, *CYP2B11*, and *CYP2D15* (NCBI entries NM_001177805.1, NM_001006652.1, and NM_001003333.1, respectively) were synthesized and cloned into the pFastBac1™ vector (Thermo Fisher Scientific) by GenScript (Piscataway, NJ, USA). Site-directed mutagenesis to obtain each POR variant was also performed by GenScript. High titer stocks containing POR-H1, each POR variant (H2 –H4), CYP2B11, and CYP2D15 recombinant baculoviruses were created using the Bac-to-Bac® Baculovirus Expression System following the manufacturer's protocols (Thermo Fisher Scientific) and as previously described[25]. Recombinant baculoviruses containing the manufacturer-supplied control expression plasmids containing β-glucuronidase (GUS) and pFastBac empty vector (PFEV) were also created. Gel electrophoresis and DNA sequencing confirmed the presence of the cDNA in recombinant baculoviruses. Sf9 shaking suspension cultures were grown in the dark at 27°C in Sf-900™ II serum-free medium (Thermo Fisher Scientific) supplemented with 5% fetal bovine serum (HyClone Laboratories, Logan, UT, USA) to a density of $1.5 \times 10^6$ cells/mL for infection.

Amplified viral stocks were titered relative to the recombinant POR-H1 baculovirus stock using a TaqMan® gene expression assay (Thermo Fisher Scientific) that measures viral Gp64 DNA concentration by QPCR as described by Hitchman *et al* [37]. This assay has been validated as an accurate and reproducible alternative to plaque assay titration. Preliminary experiments were conducted to determine the POR-H1 viral titer that resulted in maximal cytochrome c reductase activity in infected cells. The same viral titer was then used in subsequent experiments to infect cells with POR-H1, each POR variant or the GUS baculoviruses. CYP2B11 and CYP2D15 viral titers that resulted in maximal 7-benzyloxyresorufin *O-*

debenzylation and tramadol *O*-demethylation (respectively) when coexpressed with POR-H1 were also determined and used for coexpression experiments with POR-H1, each POR variant and PFEV control.

For P450 coexpression experiments, 2 μg/mL hemin (prepared by dissolving hemin in 50% ethanol and 0.2 M sodium hydroxide) was added to the culture 24 h post-infection[38]. Cells were harvested 72 h post-infection by centrifugation and washed twice with 4˚C phosphate buffered saline (pH 7.4) and stored at -80˚C until use. All protein expression experiments were repeated four independent times resulting in 4 separate batches for testing.

Sf9 microsomes were prepared as previously described [25] and stored at -80ºC until use. Microsomal protein concentrations were measured using the Pierce bicinchonic acid assay kit (Thermo Fisher Scientific). The cytochrome P450 content of microsomal preparations were also determined with the carbon monoxide P450 spectrum difference assay using an extinction coefficient ($\Delta_{\varepsilon 450-490}$) of 91,000 $M^{-1}$ $cm^{-1}$ as previously described [2, 39, 40].

## Immunoquantitation of recombinant POR, CYP2B11 and CYP2D15

Relative microsomal POR, CYP2B11, and CYP2D15 protein content were determined by immunoblotting. Microsomal fractions (2 μg of microsomal protein for POR and CYP2B11-POR co-expression experiments and 5 μg of microsomal protein for CYP2D15-POR co-expression experiments) were treated with sample reducing agent (95% Laemmli Buffer and 5% β-mercaptoethanol, Bio-Rad) and denatured at 95˚C for 10 minutes. Separation was carried out on 4–15% Mini-PROTEAN® TGX™ gels in Tris-glycine-SDS running buffer and electrophoretically transferred to polyvinylidene difluoride membranes (Bio-Rad). Blots were blocked with Tris-buffered saline with 0.05% Tween®-20 and 2% (w/v) non-fat dry milk for one hour at room temperature. Rabbit anti-human POR polyclonal antibody was from GenScript. Rabbit anti-dog CYP2B11 serum (α-PBD-2) was kindly gifted by Dr. James Halpert (School of Pharmacy, University of Connecticut, Storrs, CT, USA)[41]. Rabbit anti-human CYP2D6 polyclonal antibody was purchased from Abcam (Cambridge, MA, USA). Blots were incubated with POR, CYP2B11, or CYP2D6 antibodies at 1:10,000, 1:5,000, or 1:3,750, respectively, overnight at 4˚C. All blots were incubated with a secondary horseradish peroxidase-conjugated goat anti-rabbit antibody (Santa Cruz Biotechnology, Inc., Santa Cruz, CA, USA) at a dilution of 1:10,000 for one hour at room temperature. Proteins were detected with CLarity Western ECL chemiluminescent blotting substrate (Bio-Rad) using a ChemiDoc™ MP Imaging System with Image Lab™ software (Bio-Rad). Protein band densities from immunodetection were measured using ImageJ 1.51 software[42]. After immunodetection, blots underwent Coomassie staining as detailed by Welinder and Ekblad [43] to detect total protein. Blots were reimaged with the ChemiDoc™ MP Imaging System. The total protein staining density for each lane was analyzed with ImageJ 1.51 software with an area outside the lanes defined as background and subtracted to give a background-corrected total lane density. Protein band densities from immunodetection were normalized to the corresponding lane's background-corrected total lane density from total protein staining as described by Welinder and Ekblad [43]. Immunoblotting experiments were repeated three times using four independently generated protein preparations with results averaged. Preliminary studies were conducted using serial dilutions of recombinant POR-H1 and CYP-POR-H1 co-expressing microsomes to ensure a linear relationship between the amount of microsomal protein loaded and band intensity.

## POR enzyme activity

To assess the effects of the *POR* variants on basic enzyme quality and function, their ability to receive electrons from NADPH or donate electrons to cytochrome c was examined. The assays

were carried out as previously described[24]. For microsomes containing only recombinant POR or POR variants, 0.5 μg of total microsomal protein was used for each reaction. The final NADPH concentration was 100 μM when cytochrome c concentration was varied from 0–100 μM, and the final cytochrome c concentration was 30 μM when NADPH concentration was varied from 0–75 μM. The reaction rates were determined using the linear range of the kinetic traces and the molar extinction coefficient for reduced cytochrome c ($21.1 \text{ mM}^{-1} \text{ cm}^{-1}$). Enzyme kinetic parameters were derived by fitting a one-enzyme Michaelis-Menten model to measured enzyme activities and substrate concentrations using nonlinear regression (Sigma-Plot 13 software, Systat Software Inc., San Jose, CA, USA). Enzyme kinetic studies were conducted using four independently generated protein preparations, each assayed on three separate days and results were averaged.

To determine POR activity in microsomes co-expressed with CYP2B11 and CYP2D15, the same procedure described above was followed. Each reaction used 1 μg of total microsomal protein and final in-well concentrations of 30 μM cytochrome c and 100 μM NADPH were used. Activities were measured using four independently generated protein preparations on three separate days and results were averaged.

## CYP enzyme activities

Enzymatic activities selective for CYP2B11 or for CYP2D15 were measured using microsomes prepared from insect cells expressing the respective recombinant CYP coexpressed with POR, each POR variant, or the empty vector control. CYP2B11 activities measured included 7-benzyloxyresorufin *O*-debenzylation [44], propofol 4-hydroxylation [24], and bupropion 6-hydroxylation [25]. CYP2D15 activities were measured by tramadol *O*-demethylation [45] and dextromethorphan *O*-demethylation [24]. Preliminary experiments were conducted to confirm the linearity of metabolite formation with microsomal protein concentrations and incubation times for all activities. Final data used were normalized to total microsomal P450 content and incubation time. Eadi-Hofstee plots showed linearity for all data sets, consistent with one enzyme Michaelis-Menten enzyme kinetics. Consequently, the enzyme kinetic parameters maximal velocity ($V_{max}$) and Michaelis-Menten constant ($K_m$) were estimated by fitting the Michaelis-Menten model to enzyme reaction velocity and substrate concentration data using nonlinear regression analysis (SigmaPlot Version 13 software). Intrinsic clearance ($CL_{int}$) values were calculated by dividing the derived $V_{max}$ and $K_m$ values. All enzyme kinetic studies were conducted using four independently generated protein preparations, each assayed on three separate days and results were averaged.

7-Benzyloxyresorufin *O*-debenzylation activities were quantified as previously described by Chang and Waxman[46] with slight modifications to fit a microplate assay. A final microsomal protein concentration of 1 pmol of CYP2B11 per mL incubation volume was used with final BROD concentrations (dissolved in 0.5% acetonitrile) varying from 0.75–10 μM for kinetic studies. Resorufin formation was continuously monitored in kinetic mode at 37° for 45 minutes using the SpextraMax® i3 plate reader operated with Softmax Pro 6.3 software (Molecular Devices, San Jose, CA, USA) at excitation and emission wavelengths of 530 and 582 nm, respectively. Resorufin formation was determined using a resorufin standard curve (0–100 nM) run concurrently with samples. Reaction rates were determined from the linear range of the kinetic trace. Propofol 4-hydroxylation activities were quantified as previously described by Martinez *et al.*[24]. A final microsomal protein concentration of 3 pmol of CYP2B11 per mL incubation volume was used with propofol concentrations (dissolved in 0.5% acetonitrile) ranging from 5 to 100 μM for kinetic studies. Bupropion 6-hydroxylation activities were quantified as previously described [25]. A final microsomal protein concentration of 2 pmol of

CYP2B11 per mL incubation volume was used and final bupropion concentrations (dissolved in 0.5% acetonitrile) ranged from 10 to 500 μM for kinetic studies.

Tramadol *O*-demethylation activities by recombinant CYP2D15 co-expressed with POR, POR variants, or the empty vector control were assessed and quantified as previously described by Perez *et al.*[45]. A final concentration of 2.5 pmol of CYP2D15 per mL incubation volume was used and final tramadol concentrations (dissolved in 0.5% acetonitrile) ranged from 2–100 μM for kinetic studies. Dextromethorphan demethylation activities were assessed and quantified as previously described [24]. A final concentration of 2.5 pmol of CYP2D15 per mL incubation volume was used and final dextromethorphan concentrations (dissolved in 0.5% acetonitrile) ranged from 0.5 to 25 μM for kinetic studies.

## Statistical analysis

Statistical analyses were performed using Sigma Plot 13 software. Haplotype frequencies between sighthound and non-sighthound breed groups were evaluated using the Mann-Whitney *U* test. The influence of haplotype on protein abundance, kinetic parameters, and activities of recombinant microsomes was evaluated by ANOVA with *post-hoc* pairwise testing using the Holm-Sidak multiple comparison test. If data did not meet parametric testing prerequisites (normality and equal variance between groups), it was evaluated using ANOVA on ranks, followed by *a post-hoc* Tukey test. The relationships between POR and CYP activities and between substrates were determined by calculating Spearman's correlation coefficients. For all statistical tests, a P-value of < 0.05 was considered statistically significant.

## Results

### Identification of canine *POR* genetic polymorphisms

Whole transcriptome sequencing (RNA-seq) data using liver tissue from five NGA-registered greyhounds (slow CYP2B11 metabolizer phenotype) and five beagles (fast CYP2B11 metabolizer phenotype) reported in our previous study of *CYP2B11* pharmacogenomics [25] were evaluated for possible *POR* mRNA splicing variation and cDNA sequence polymorphisms that differed between the two breeds. Compared with the canonical *POR* cDNA sequence (NM 00177805), no RNA splice variants were identified in any of the liver samples examined. However, two nonsynonymous SNPs (cSNPs) in exon 9 (c.943 G/C, Glu315Gln) and exon 13 (c.1710 C/G, Asp570Glu) were discovered in the greyhound liver samples but not in any of the beagle liver samples (S3 Table). DNA extracted from the 5 liver samples and buccal samples from an additional 8 (unrelated) greyhounds were then sequenced to confirm the presence of the exon 9 and 13 cSNPs and identify any additional coding variants. As shown in S3 Table, 11 of 13 greyhound DNA samples tested had one (2 of 11) or both (9 of 11) of these nonsynonymous SNPs. Although 3 synonymous SNPs were found in exons 3, 5, and 13, no additional cSNPs were found in any of the 15 *POR* exons examined.

### *POR* cSNP population frequency and breed heterogeneity

Canine population genotype frequencies and breed-associated heterogeneity of the *POR* c.943 G/C and c.1710 C/G cSNPs were evaluated by genotyping DNA samples collected from 68 different breeds. At least 10 dogs per breed were genotyped. Breeds sampled included 21 sighthound breeds, 47 other dog breeds, and as well as mixed-breed dogs (2,286 dogs in total). DNA samples from greyhounds (n = 258) included those from 196 NGA-registered greyhounds (bred for racing) and from 62 AKC-registered greyhounds (bred for conformation).

Genotype frequency data for the c.943 G/C and c.1710 C/G cSNPs in all dogs are given in S4 Table. Four haplotypes could be inferred from these data, designated *POR*-H1 (reference), *POR*-H2 (c.1710G), *POR*-H3 (c.943C and c.1710G), and *POR*-H4 (c.943C). *POR* haplotype frequencies for sighthounds and other dog breeds with at least one variant haplotype are given in Table 1, while data for all breeds are shown in S1 Fig. *POR*-H2 was found in 10 of the 21 sighthound breeds (48%) and in 12 of the 47 (26%) other dog breeds. Rottweilers had the highest *POR*-H2 frequency (51%) among all breeds. The average (± SE) *POR*-H2 frequencies for sighthounds and other dog breeds were similar (P = 0.12, Mann-Whitney *U* test) at 2.8 ± 1.1% and 2.7 ± 1.2%, respectively. Conversely, the *POR*-H3 haplotype was relatively restricted to sighthound breeds, found in 8 of 19 (37%) sighthound breeds, but only in two of 45 non-sighthound breeds (4%). The average *POR*-H3 haplotype frequency in sighthound breeds (6.0 ± 2.3%) was 50-fold higher (P < 0.0001, Mann-Whitney *U* test) than that in other breeds (0.12 ± 0.10%).

**Table 1.  *POR* haplotype frequency heterogeneity across dog breeds[c].**

| | | *POR* haplotype frequency (%) | | |
|---|---|---|---|---|
| | N dogs tested | *POR*-H2 | *POR*-H3 | *POR*-H4 |
| **SIGHTHOUND BREEDS** | | | | |
| Greyhound (NGA) [a] | 180 | 4.7 | 35 | 0.3 |
| Scottish Deerhound | 137 | 0.7 | 35 | 0.7 |
| Galgo Español | 25 | 4.0 | 16 | 0 |
| Cirneco dell'Etna | 15 | 0 | 11 | 0 |
| Greyhound (AKC) [b] | 59 | 0 | 10 | 0 |
| Portuguese Podengo | 12 | 4.2 | 4.2 | 0 |
| Saluki | 14 | 18 | 3.6 | 0 |
| Afghan | 27 | 3.7 | 0 | 0 |
| Anatolian Shepherd | 51 | 14 | 0 | 0 |
| Basenji | 25 | 2.0 | 0 | 0 |
| Borzoi | 51 | 14 | 0 | 0 |
| Italian Greyhound | 54 | 0.9 | 0 | 0 |
| **OTHER BREEDS** | | | | |
| Rottweiler | 41 | 51 | 0 | 0 |
| Chow Chow | 36 | 19 | 0 | 0 |
| Doberman Pinscher | 35 | 17 | 0 | 0 |
| American Staffordshire Terrier | 21 | 7.1 | 0 | 0 |
| Chesapeake Bay Retriever | 16 | 6.3 | 0 | 0 |
| Saint Bernard | 23 | 4.3 | 0 | 0 |
| Jack Russell Terrier | 12 | 4.2 | 0 | 0 |
| Bernese Mountain Dog | 27 | 3.7 | 0 | 0 |
| Chihuahua | 14 | 3.6 | 0 | 0 |
| Pitbull | 33 | 3.0 | 0 | 0 |
| Border Collie | 67 | 0.7 | 4 | 0 |
| Labrador Retriever | 61 | 0 | 0 | 1 |
| Mixed Breed Dogs | 147 | 4.4 | 1 | 0 |

[a] National Greyhound Association (NGA).

[b] American Kennel Club (AKC).

[c] Shown are the variant haplotype frequencies for all dog breeds tested that had at least one of the variant haplotypes. A listing of all other dog breeds that were evaluated in which no variant haplotypes were found are given in S1 Fig.

Differences in *POR* haplotype allele frequencies were also observed between NGA-registered greyhounds and AKC-registered greyhounds. The *POR*-H2 allele frequency in NGA-registered greyhounds was 4.7%, while none of the AKC-registered greyhounds had this allele. NGA-registered greyhounds also had more than three-fold higher *POR*-H3 allele frequency (35%) than AKC-registered greyhounds (10%). NGA-registered greyhounds and Scottish deerhounds had the highest *POR*-H3 allele frequencies (35% for both) among all breeds surveyed. *POR*-H4 was rare. This haplotype was found in only two of 138 Scottish deerhounds (0.6% allele frequency), one of 196 NGA-registered greyhounds (0.3% allele frequency), one of 61 Labrador retrievers (0.8% allele frequency), and one of 67 border collies (0.6% allele frequency). All five dogs were heterozygous for the *POR*-H4 allele.

### *In silico* analysis of *POR* amino acid variants

Multiple sequence alignments of dog POR with homologs from other species showed that the 315 glutamate residue was largely conserved, indicating that this residue may be significant to the structure or function of the protein (Fig 1A). Zebrafish and fruit flies possessed an aspartate in place of the glutamate at 315, but both are similar positively charged amino acids, suggesting a requirement for this property. Consequently, the substitution in dogs at position 315 with glutamine, a polar neutral amino acid, is likely a disruptive change. On the other hand, the 570 aspartate residue was not conserved within or outside of mammals, indicating that this residue may not be important for the structure or function of the protein (Fig 1B). Furthermore, as mentioned previously, the substitution of a glutamate for the aspartate at 570 in dogs is unlikely to be a disruptive change. These results were confirmed quantitatively by PolyPhen-2 analysis, which indicated that the Glu315Gln amino acid change was probably damaging, with a score of 0.971 (sensitivity: 0.77; specificity: 0.96), whereas Asp570Glu was predicted to be benign, with a score of 0.000 (sensitivity: 1:00; specificity: 0.00).

A three-dimensional homology model of canine POR was then built using x-ray crystal structures of human (PBD: 3QE2, 3QFC) and rat (PBD: 4YAL, 5URD, 1JA1) POR. The best (highest scoring) regions of each of the individual human and rat structures were used to generate a hybrid canine POR model that optimized residue coverage and enhanced model accuracy beyond that afforded by each of the individual structures. The final canine POR model

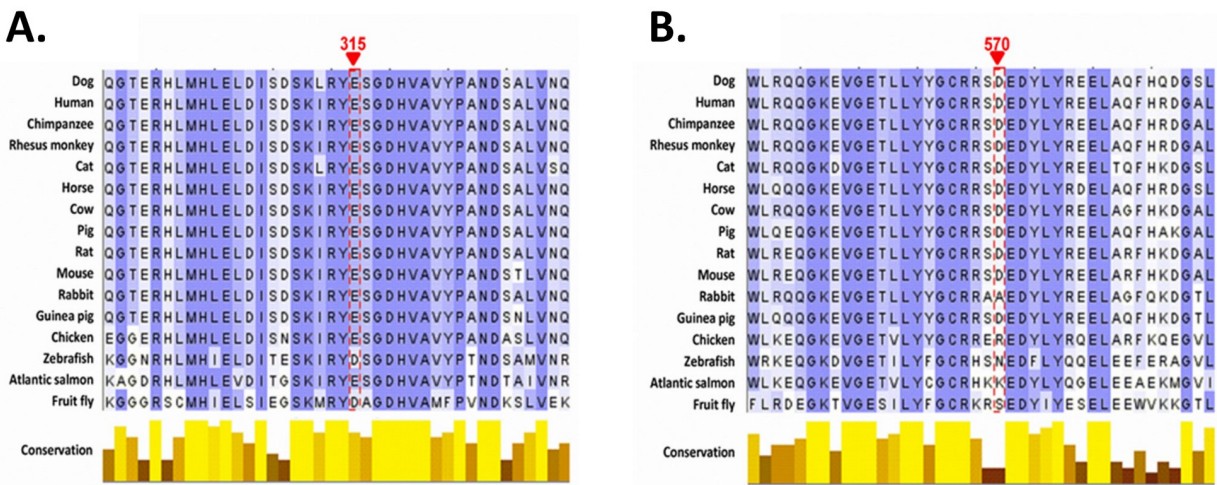

**Fig 1. Sequence conservation of POR.** Multiple sequence alignment of POR amino acid sequences showing (A) the glutamate 315 residue and (B) the aspartate 570 residues.

was then used to understand the structural basis of changes caused by the Glu315Gln and Asp570Glu mutations. The 3-dimensional model obtained for WT canine POR is shown in Fig 2. The Glu315 residue is located within the predicted ferredoxin reductase-type FAD-binding domain, based on computer-generated annotations derived from UniProt sequences P00388 (NCPR_Rat), P16435 (NCPR_Human), and P16603 (NCPR_Yeast). The replacement of Glu315 by Gln was predicted to disrupt the interaction with Arg519, perturbing the structural stability of the enzyme (Fig 3). The empirical protein design forcefield FoldX predicted a difference in free energy of the mutation Glu315Gln to be 1.6 kcal/mol, indicating a destabilizing effect of the mutation. The same destabilizing effect was predicted by DynaMut (-0.151 kcal/mol), SDM (-0.69 kcal/mol) and DUET (-1.19 kcal/mol). A calculation of vibrational entropy energy difference between WT and Glu315Gln mutant proteins showed a ΔΔSvib of 0.16 kcal$^{-1}$.mol.K$^{-1}$ indicating an increase in molecular flexibility of the mutant protein (Fig 4).

Disruption to the structural stability of the enzyme could potentially lead to a change in interaction with cytochrome P450 partner proteins and impact the activities of cytochromes

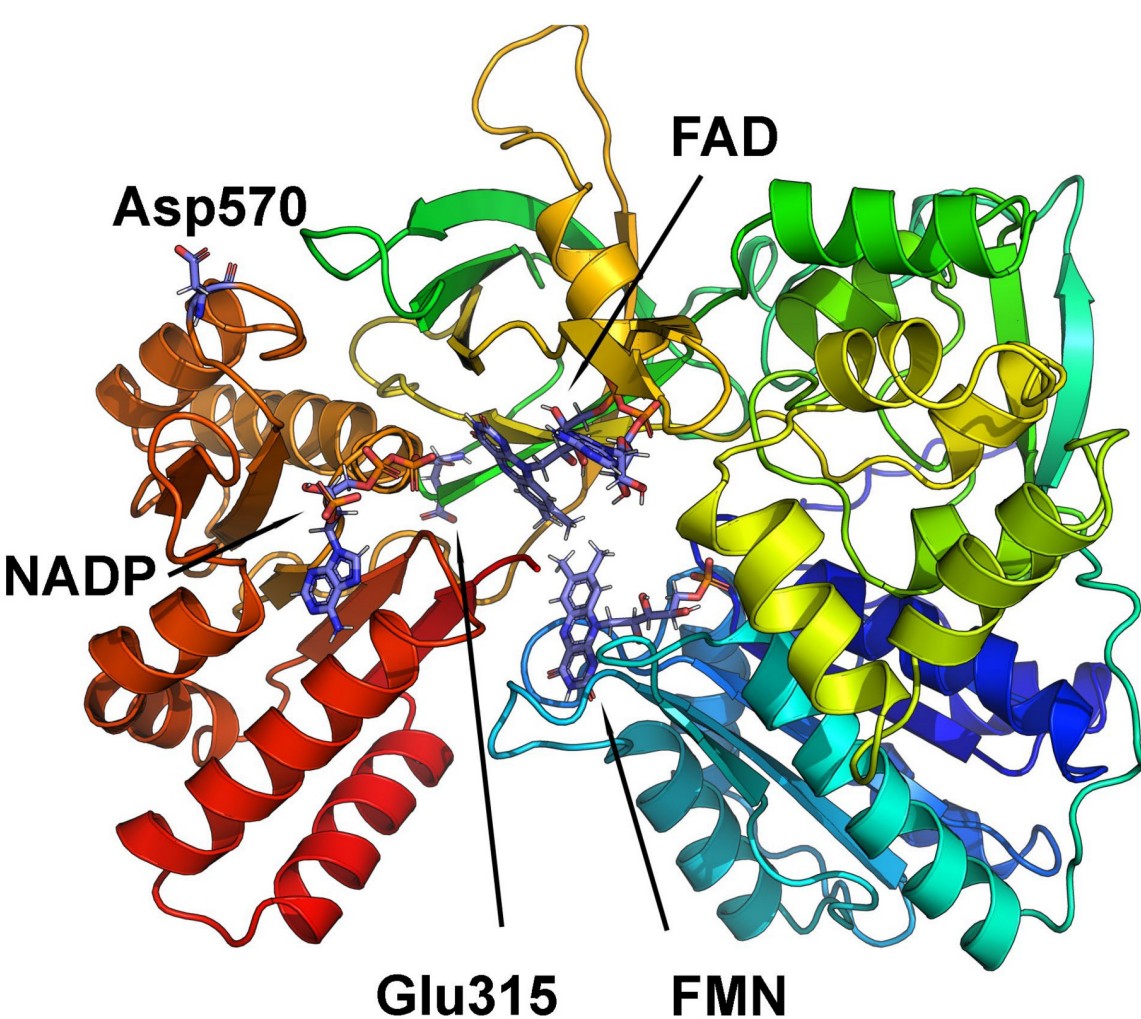

**Fig 2. Canine POR 3D model showing POR variants.** A 3D model of canine POR is displayed as a ribbons model showing the positions of the Glu315 and Asp570 residues. Structural models of dog POR are based on known 3D structures of the human and rat POR protein as described in methods. Protein is colored in rainbow with blue at N-terminus and red at C-terminus. Cofactors (NADP, FAD, and FMN) and amino acids Glu315 and Asp570 are shown as stick models.

P450 that require POR as their redox partner. Based on the computer-predicted annotation of the canine POR in UniProt, the Asp570 residue is located in a loop within the NADP$^+$ binding region (Fig 2). Stability calculations predicted that the replacement of Asp570 by Glu would not disrupt atomic interactions within POR (Fig 3). Consequently, no disruption to the structural stability of the enzyme was predicted. All prediction methods suggested a slight increase in protein stability (FoldX -0.14 kcal/mol, SDM 0.01 kcal/mol, DUET 0.014 kcal/mol and DynaMut 0.249 kcal/mol). Calculation of vibrational entropy energy difference between WT and Asp570Glu mutant proteins showed a ΔΔSvib of -0.099 kcal$^{-1}$.mol.K$^{-1}$ suggesting a slight decrease in molecular flexibility (Fig 4).

## Effect of POR variants on POR, CYP2B11, and CYP2D15 protein expression

Reference canine POR (POR-H1) and the three protein sequence variants (POR-H2: Asp570Glu; POR-H3: Glu315Gln/Asp570Glu; POR-H4: Glu315Gln) were successfully

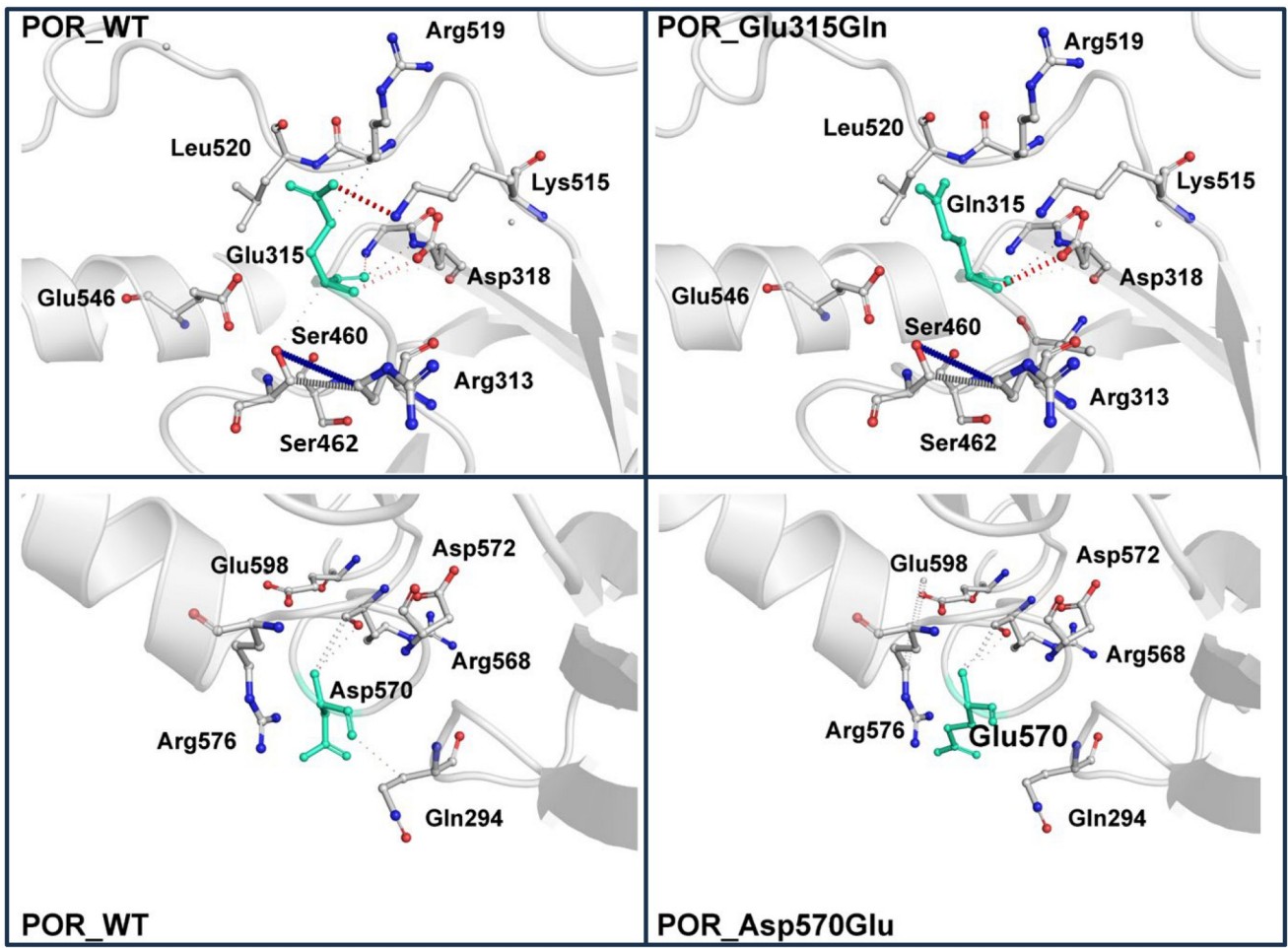

**Fig 3. A closeup of the WT and mutant canine POR proteins showing the molecular interactions within POR.** Top left: In the WT POR Glu315 forms a salt bridge with Arg519 and has further interactions with Lys515 that are responsible for structural stability. Top right: Upon mutation of Glu315 to Gln, while some molecular interaction with Lys515 was still present, the salt bridge with Arg519 was broken, suggesting an impact on structural stability. Bottom left: The Asp570 is involved in molecular interactions with Arg568. Bottom right: Upon mutation to Glu570, the interactions with Arg568 are still present indicating no impact on structural stability.

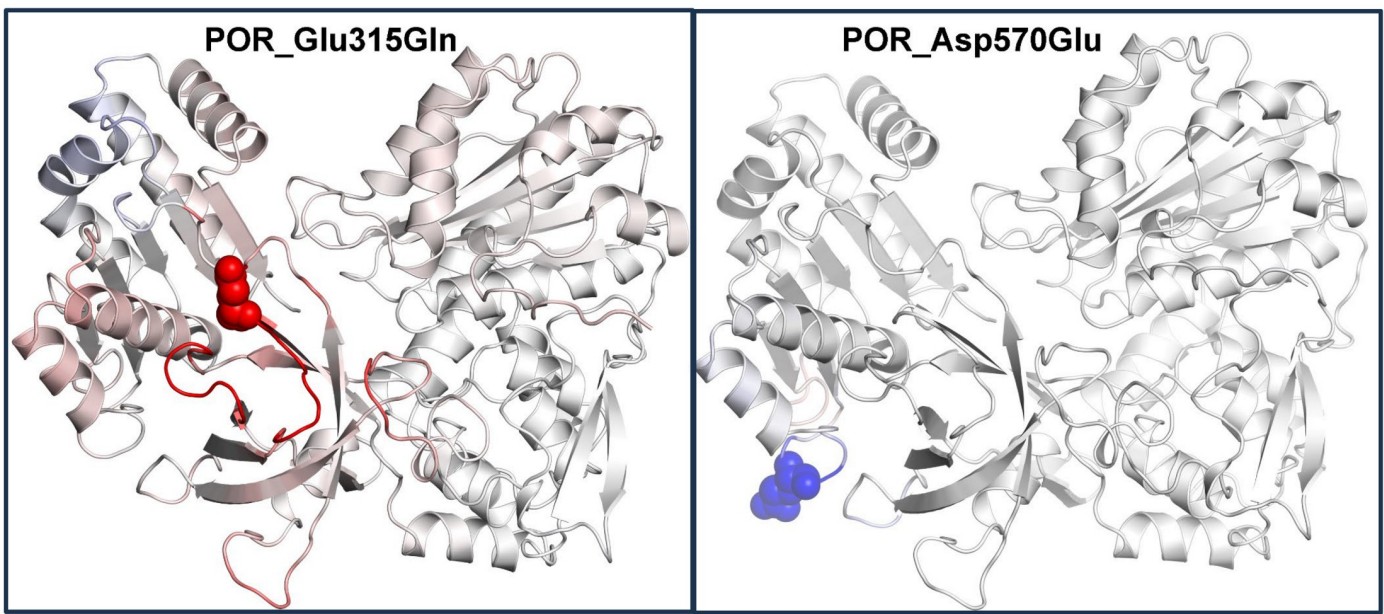

**Fig 4. Stability and flexibility analysis of the mutated POR structures compared to WT POR.** Left panel: An increased flexibility in Glu315Gln POR indicating decreased stability with was supported by differential free energy calculations. Right panel: Decreased flexibility due to Asp570Glu mutation in canine POR, which was supported by increased stability of POR structure due to Asp570Glu mutation in differential free energy analysis using DynaMut, SDM and DUET as well as FoldX predictions. Protein flexibility, compared to WT canine POR, is colored red (increased flexibility) and blue (reduced flexibility).

expressed in Sf9 insect cells using the Bac-to-Bac® system. CYP2B11 and CYP2D15 were also successfully co-expressed with POR and each POR variant using the same expression system. The relative expression of POR, CYP2B11, and CYP2D15 proteins in the Sf9 microsomes was determined by immunoblotting. As shown in Fig 5, there were no differences in immuno-quantified POR protein concentration between reference POR and each POR variant when expressed alone or with either CYP2B11 or CYP2D15 ($P > 0.05$, ANOVA). Furthermore, there were no differences in immunoquantified CYP2B11 or CYP2D15 concentration when these P450 enzymes were co-expressed with each of the POR variants ($P > 0.05$, ANOVA). POR protein was not detected in immunoblots of Sf9 cells infected with the pFastBac1 empty vector (PFEV) negative control or β-glucuronidase (GUS) negative control instead of POR.

### Effect of POR variants on cytochrome c reductase activities

Cytochrome c reductase enzyme kinetics were evaluated to assess the effect of the POR amino acid variant substitutions on basic enzyme quality, general catalytic efficiency, and electron transfer within POR. Experiments were first conducted using POR expressed alone (without P450) kinetic parameters determined using fixed NADPH concentration and varied cytochrome c concentrations or fixed cytochrome c concentration and varied NADPH concentration. Enzyme kinetic plots from these experiments are shown in Fig 6, and the derived enzyme kinetic parameters are given in Table 2. As shown in Table 2, although there was a trend for lower mean $V_{max}$ and $CL_{int}$ for the H3 and H4 variants compared to the reference H1 POR for both cytochrome c and NADPH kinetics, these differences did not reach statistical significance ($P > 0.05$, ANOVA). Cytochrome c $K_m$ values were also not different between variants and reference POR. However, the mean (± SD) NADPH $K_m$ value for H4 (2.5 ± 0.3 μM) was significantly lower than the $K_m$ for H1 (3.2 ± 0.2 μM) ($P = 0.045$, ANOVA with Holm-Sidak test).

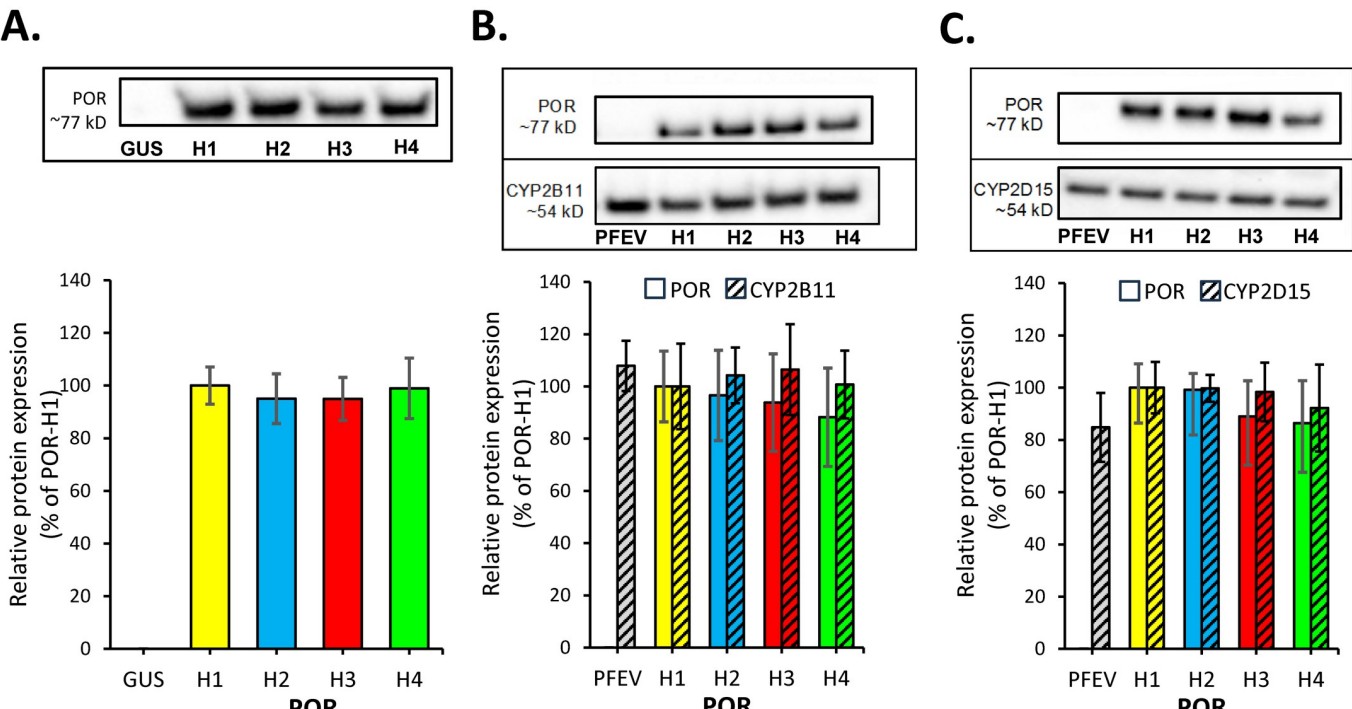

**Fig 5. Effect of POR mutations on POR, CYP2B11 and CYP2D15 protein expression in Sf9 cells.** Immunoblotting was performed using microsomes from Sf9 cells expressing wild-type POR (H1) or each POR variant (H2, H3 and H4) either alone (A), or co-expressed with CYP2B11 (B), or with CYP2D15 (C). Shown are the mean (± SD) immunodetectable protein content of POR (solid bars), CYP2B11 (striped bars in B) and CYP2D15 (striped bar in C) from 4 independent experiments with 3 blots prepared from each experiment. Data were normalized to total loaded protein content and expressed as a percentage of the POR-H1expressing microsomes. Shown above each bar chart are examples of blots used to derive these data (full length blots shown in S2 Fig). B-glucuronidase (GUS) was substituted for POR as a negative control in A, while pFastBac1 empty vector (PFEV) was substituted for POR as a negative control in B and C. No significant differences in POR expression or CYP co-expression between POR variants were identified by ANOVA (P > 0.05).

Cytochrome c reduction activity was also measured in microsomes co-expressing CYP2B11 and CYP2D15 using a fixed concentration of NADPH (100 μM) and cytochrome c (30 μM) (Fig 7). For both CYP2B11 and CYP2D15 experiments, co-expression with empty vector control instead of POR resulted in very low mean (± SD) POR activities of 8.5 ± 2.4 and 6.8 ± 1.4 nmol cytochrome c reduced / min / mg microsomal protein (respectively), which were less than 5% of POR activities measured in microsomes coexpressing POR-H1 with CYP2B11 or CYP2D15. In CYP2B11 co-expression experiments, POR-H4 was found to have significantly lower POR activities by about 60% compared with POR-H1 (P = 0.013, ANOVA with Holm-Sidak test) (Fig 7A). However, for CYP2D15 co-expression experiments, no significant differences in POR activities were found between POR-H1 and all other POR variants (P > 0.05, ANOVA) (Fig 7B).

## Effect of POR variants on CYP2B11 enzyme kinetics

CYP2B11 enzyme kinetics were evaluated in recombinant microsomes coexpressing CYP2B11 with POR-H1, each POR variant, or with the empty vector control. CYP2B11 activity probes that were evaluated included 7-benzyloxyresorufin (BROD), propofol, and bupropion. Fig 8 shows the enzyme kinetic plots for BROD (Fig 8A and 8B), propofol (Fig 8C and 8D), and bupropion (Fig 8E and 8F) oxidation. Derived enzyme kinetic parameter estimates are summarized in Table 3. For all 3 substrates tested, compared with POR-H1, co-expression with

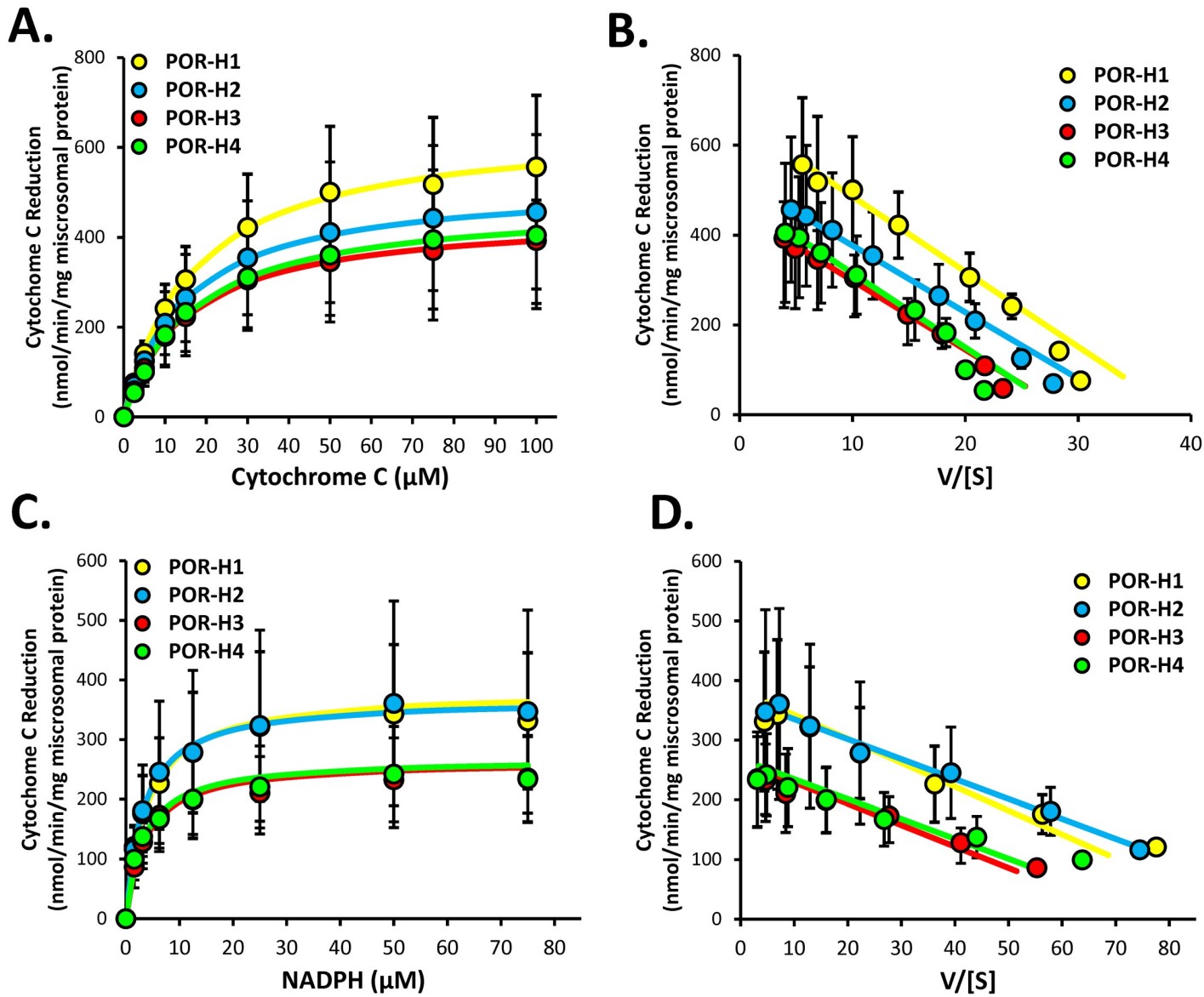

**Fig 6. Effect of POR mutations on cytochrome c reduction by POR expressed in Sf9 cells.** Shown are Michaelis-Menten enzyme kinetic plots of rates of microsomal cytochrome c reduction by POR-H1 and each POR variant (H2-H4) when (A) cytochrome c concentration is varied and (C) NADPH concentration is varied. Also shown are Eadie-Hofstee plots of these same data (B and D, respectively). Data points represent the mean (± SD) activities measured using four independently generated protein preparations, each assayed on three separate days. The curves represent the kinetic model of best fit to each data set. Enzyme kinetic parameter estimates are given in Table 2.

POR-H3 or POR-H4 significantly reduced CYP2B11 $V_{max}$ and $CL_{int}$ values, without affecting $K_m$ values (P < 0.05, ANOVA with Holm-Sidak test). CYP2B11 $CL_{int}$ values for POR-H3 averaged 34%, 39%, and 37% lower that POR-H1 for oxidation of BROD, propofol and bupropion, respectively. The effect of POR-H4 on CYP2B11 $CL_{int}$ was more substantial, averaging 65%, 72%, and 69% lower values compared with POR-H1for oxidation of BROD, propofol and bupropion, respectively. CYP2B11 $CL_{int}$ values for POR-H2 coexpressed microsomes were not different from POR-H1 microsomes regardless of substrate testes. As expected, CYP2B11 activities were highly dependent on POR coexpression, with $CL_{int}$ values for empty vector control infected microsomes averaging less than 3% of POR-H1 microsomes.

**Table 2. Michaelis-Menten enzyme kinetic parameters (mean ± SD) determined for cytochrome c reduction by recombinant POR-H1 and POR variants (H2, H3 and H4)[a].**

| POR variant | $V_{max}$ (nmol/min/mg) | $K_m$ (µM) | $CL_{int}$ (mL/min/mg) |
|---|---|---|---|
| Cytochrome c enzyme kinetics [b] | | | |
| H1 | 653 ± 205 | 16.7 ± 2.0 | 38 ± 7 |
| H2 | 525 ± 195 | 14.8 ± 1.4 | 35 ± 12 |
| H3 | 453 ± 104 | 15.4 ± 1.3 | 29 ± 6 |
| H4 | 480 ± 188 | 16.5 ± 1.6 | 29 ± 10 |
| NADPH enzyme kinetics [c] | | | |
| H1 | 356 ± 128 | 3.2 ± 0.2 | 110 ± 35 |
| H2 | 370 ± 185 | 3.2 ± 0.5 | 111 ± 41 |
| H3 | 242 ± 74 | 2.7 ± 0.3 | 89 ± 24 |
| H4 | 244 ± 78 | 2.5 ± 0.2* | 99 ± 33 |

[a] Concentration and activity data used to derive these estimates and curves of best fit are shown in Fig 6.

[b] Cytochrome c kinetics were determined by holding NADPH concentrations constant and varying cytochrome c concentrations.

[c] NADPH kinetics were determined by holding cytochrome c concentrations constant and varying NADPH concentrations.

*$P < 0.05$ compared with POR-H1; ANOVA with Holm-Sidak test.

$V_{max}$ values for 7-benzyloxyresorufin *O*-debenzylation, propofol hydroxylase, and bupropion hydroxylase for each POR variant were all strongly correlated with one another (Rs ≥ 0.87, $P < 0.001$, Spearman's correlation) (S3A–S3C Fig). Furthermore, $V_{max}$ values for 7-benzyloxyresorufin *O*-debenzylation, propofol hydroxylase, and bupropion hydroxylase for each POR variant were also highly correlated with cytochrome c (POR) activity (Rs ≥ 0.73, $P < 0.001$) (S3D–S3F Fig).

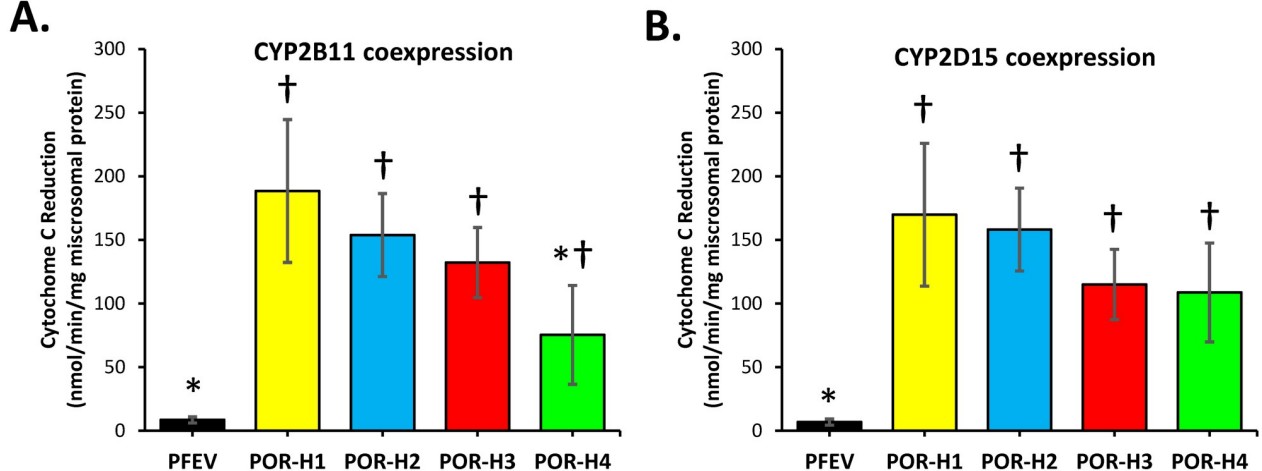

**Fig 7.** Effect of POR mutations on cytochrome c reduction by POR when co-expressed with (A) CYP2B11 or (B) CYP2D15 in Sf9 cells. Each bar represents the mean (± SD) microsomal cytochrome c activities measured using four independently generated protein preparations, each assayed on three separate days. pFastBac1 empty vector (PFEV) was substituted for POR as a negative control. *$P < 0.05$ in comparison with POR-H1, †$P < 0.05$ in comparison with PFEV. Statistical significance was evaluated by ANOVA with *post-hoc* multiple comparison Holm-Sidak test.

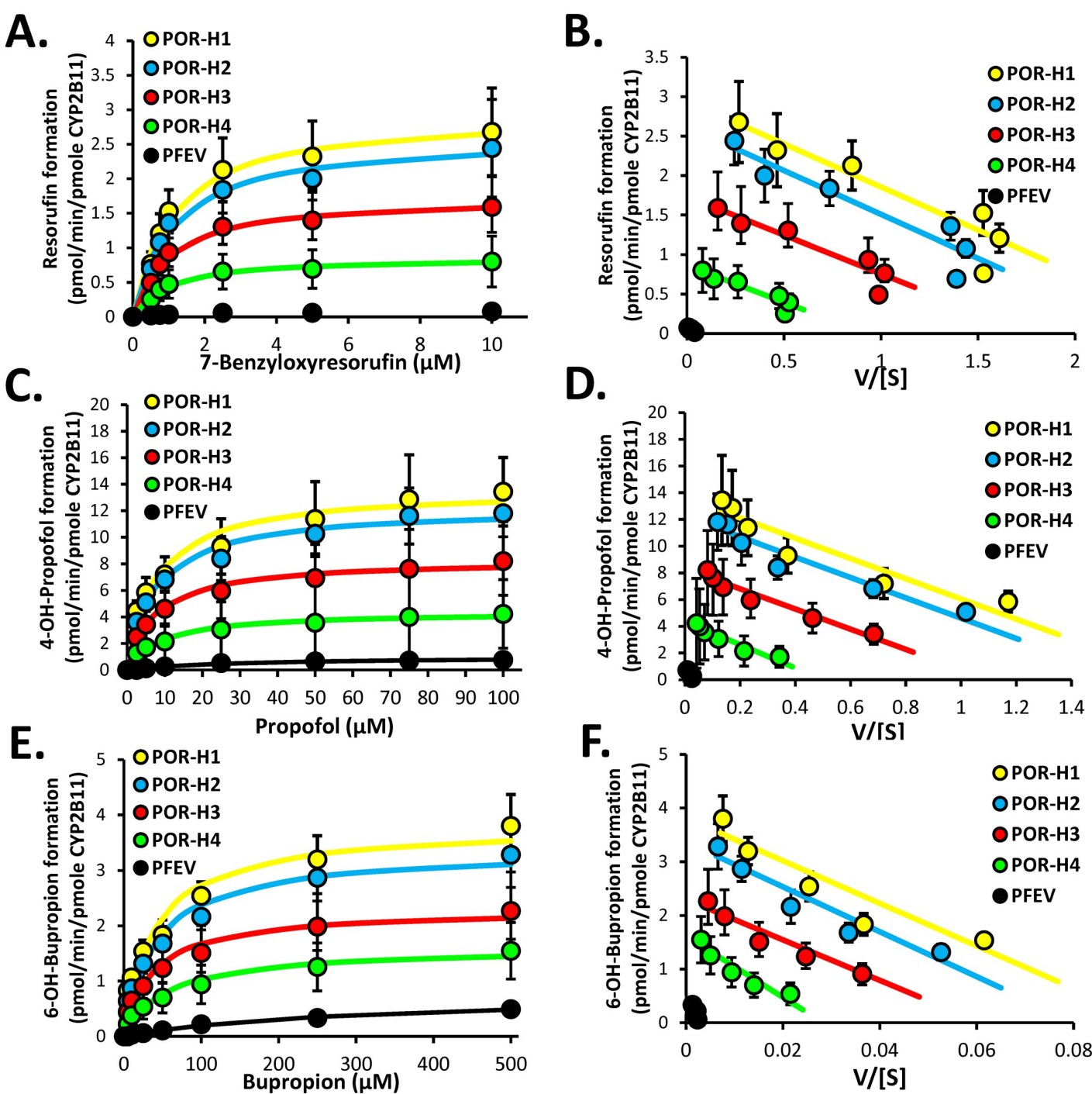

**Fig 8. Effect of POR mutations on CYP2B11 enzyme function when co-expressed with POR in Sf9 cells.** Shown are Michaelis-Menten enzyme kinetic plots of (A) resorufin, (C) 4-hydroxypropofol, and (E) 6-hydroxybupropion formation by CYP2B11 co-expressed with POR-H1, each POR variant (H2-H4), or pFastBac1 empty vector (PFEV) negative control in Sf9 microsomes. Also shown are Eadie-Hofstee plots of these same data (B, D, and F, respectively). Data points represent the mean (± SD) activities measured using four independently generated protein preparations, each assayed on three separate days. The curves represent the kinetic model of best fit to each data set. Enzyme kinetic parameter estimates are given in Table 3.

**Table 3. Michaelis-Menten enzyme kinetic parameters (mean ± SD) for oxidation of 7-benzyloxyresorufin, propofol, and bupropion by CYP2B11 coexpressed with recombinant POR-H1, POR variants (H2, H3 or H4) or pFastBac1 empty vector (PFEV) control[a].**

| POR Variant | $V_{max}$ (pmol/min/pmol P450) | $K_m$ (µM) | $CL_{int}$ (µL/min/pmol P450) |
|---|---|---|---|
| 7-Benzyloxyresorufin *O*-Debenzylation | | | |
| PFEV | 0.089 ± 0.036* | 1.66 ± 0.46* | 0.054 ± 0.027* |
| H1 | 2.95 ± 0.69[†] | 1.09 ± 0.13[†] | 2.70 ± 0.60[†] |
| H2 | 2.64 ± 0.69[†] | 1.12 ± 0.28[†] | 2.38 ± 0.34[†] |
| H3 | 1.74 ± 0.50*[†] | 1.02 ± 0.15[†] | 1.78 ± 0.80*[†] |
| H4 | 0.75 ± 0.33*[†] | 0.97 ± 0.26[†] | 0.81 ± 0.35*[†] |
| Propofol Hydroxylation | | | |
| PFEV | 0.96 ± 0.30* | 22.6 ± 2.51* | 0.042 ± 0.0* |
| H1 | 13.6 ± 3.3[†] | 7.49 ± 1.63[†] | 1.83 ± 0.30[†] |
| H2 | 12.2 ± 2.2[†] | 7.56 ± 1.37[†] | 1.62 ± 0.18[†] |
| H3 | 8.33 ± 3.18*[†] | 7.43 ± 1.10[†] | 1.11 ± 0.3)*[†] |
| H4 | 4.62 ± 1.42*[†] | 11.1 ± 5.37[†] | 0.51 ± 0.3)*[†] |
| Bupropion Hydroxylation | | | |
| PFEV | 0.80 ± 0.23* | 340 ±150* | 0.003 ± 0.002* |
| H1 | 3.82 ± 0.59[†] | 39.5 ± 4.87[†] | 0.097 ± 0.009[†] |
| H2 | 3.38 ± 0.57[†] | 41.8 ± 3.87[†] | 0.081 ± 0.009[†] |
| H3 | 2.31 ± 0.72*[†] | 37.9 ± 6.28[†] | 0.061 ± 0.018*[†] |
| H4 | 1.62 ± 0.47*[†] | 63.1 ± 21.5[†] | 0.029 ± 0.016*[†] |

[a] Concentration and activity data used to derive these estimates and curves of best fit are shown in Fig 8.

*P<0.05 compared with POR-H1; ANOVA with Holm-Sidak test.

[†]P < 0.05 compared with PFEV; ANOVA with Holm-Sidak test.

## Effect of POR variants on CYP2D15 enzyme kinetics

The effects of POR variants on CYP2D15 activities were evaluated using recombinant microsomes co-expressed with CYP2D15 and POR-H1, POR variants, or the empty vector control. Tramadol and dextromethorphan, two probe drugs for CY2D15 activity, were used as substrates in the kinetic studies to elucidate the effects of POR variants on CYP2D15 catalytic activities. Fig 9 shows the enzyme kinetic plots for tramadol and dextromethorphan metabolism, and the kinetic parameters are summarized in Table 4. Unlike CYP2B11, co-expression of CYP2D15 with POR variants did not significantly alter CYP2D15 activities when compared to co-expression with POR-H1. $K_m$, $V_{max}$, and $CL_{int}$ values for each POR variant compared to POR-H1 were not statistically different for any substrate tested (P > 0.05, ANOVA). However, co-expression of CYP2D15 with the empty vector control significantly reduced the catalytic efficiency of CYP2D15 by more than 91% compared to co-expression with POR-H1, indicating that CYP2D15 activity was highly dependent on the presence of recombinant POR (P < 0.001 ANOVA).

$V_{max}$ values for tramadol *O*-demethylation and dextromethorphan demethylation were strongly correlated (Rs = 0.95, P < 0.001, Spearman's correlation) (S4A Fig). However, $V_{max}$ values for tramadol *O*-demethylation and dextromethorphan demethylation were only moderately correlated with cytochrome c (POR) activities (Rs = 0.60, P = 0.005; and Rs = 0.58, P = 0.008, respectively) (S4B and S4C Fig).

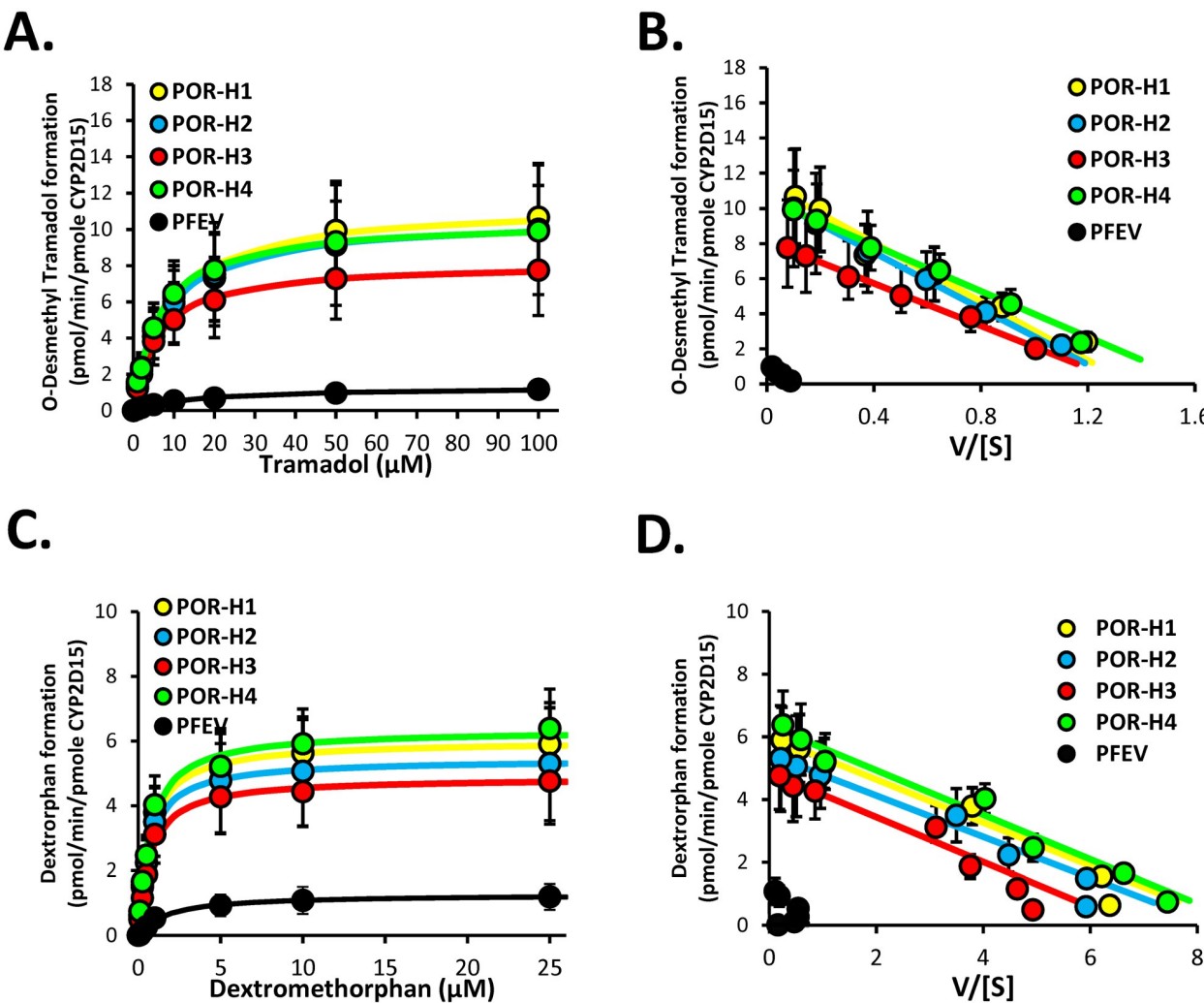

**Fig 9. Effect of POR mutations on CYP2D15 enzyme function when co-expressed with POR in Sf9 cells.** Shown are Michaelis-Menten enzyme kinetic plots of (A) *O*-desmethyl tramadol and (C) dextrorphan formation by CYP2D15 co-expressed with POR-H1, each POR variant (H2-H4) or pFastBac1 empty vector (PFEV) negative control in Sf9 microsomes. Also shown are Eadie-Hofstee plots of these same data (B and D, respectively). Data points represent the mean (± SD) activities measured using four independently generated protein preparations, each assayed on three separate days. The curves represent the kinetic model of best fit to each data set. Enzyme kinetic parameter estimates are given in Table 4.

## Discussion

We previously determined that drug metabolism mediated by CYP2B11 (but not by other CYPs) is substantially reduced in greyhounds compared with other dog breeds [25]. We identified several SNPs in the CYP2B11 gene 3'-untranslated region that substantially reduced CYP2B11 expression *in vitro*. However, the allele frequencies of these variants in greyhounds were too low to fully explain reduced CYP2B11-mediated metabolism observed in the breed. Thus, we re-examined RNA-seq data from our previous study of greyhound (slow metabolizer) and beagle (fast metabolizer) livers to identify other potential genetic causes of breed-related CYP2B11 deficiency. We focused on the *POR* gene since it encodes the obligate electron donor for all microsomal CYPs, including CYP2B11. This analysis identified two cSNPs (c.943 G/C and c.1710 C/G) in the *POR* gene that were present in all greyhound livers but

**Table 4. Michaelis-Menten enzyme kinetic parameters for oxidation of tramadol and dextromethorphan by CYP2D15 coexpressed with recombinant POR-H1, POR variants (H2, H3 or H4) or pFastBac1 empty vector (PFEV) control[a].**

| POR Haplotype | $V_{max}$ (pmol/min/pmol P450) | $K_m$ (μM) | $CL_{int}$ (μL/min/pmol P450) |
|---|---|---|---|
| | Tramadol *O*-Demethylation | | |
| PFEV | 1.32 ± 0.54[*] | 15.0 ± 3.8[*] | 0.085 ± 0.026[*] |
| H1 | 11.38 ± 2.98[†] | 8.50 ± 0.72[†] | 1.37 ± 0.47[†] |
| H2 | 10.69 ± 3.82[†] | 8.20 ± 0.87[†] | 1.34 ± 0.55[†] |
| H3 | 8.17 ± 2.74[†] | 6.59 ± 1.98[†] | 1.37 ± 0.69[†] |
| H4 | 10.52 ± 2.75[†] | 6.28 ± 1.21[†] | 1.64 ± 0.31[†] |
| | Dextromethorphan *O*-Demethylation | | |
| PFEV | 1.37 ± 0.32[*] | 1.79 ± 0.18[*] | 0.79 ± 0.27[*] |
| H1 | 6.01 ± 1.13[†] | 0.697 ± 0.068[†] | 8.79 ± 2.45[†] |
| H2 | 5.44 ± 1.85[†] | 0.664 ± 0.073[†] | 8.30 ± 2.85[†] |
| H3 | 4.87 ± 1.91[††] | 0.707 ± 0.073[†] | 6.84 ± 2.33[†] |
| H4 | 6.35 ± 1.21[†] | 0.716 ± 0.047[†] | 8.97 ± 2.14[†] |

[a] Concentration and activity data used to derive these estimates and curves of best fit are shown in Fig 9.

[*]P<0.05 compared with POR-H1; ANOVA with Holm-Sidak test.

[†]P < 0.05 compared with PFEV; ANOVA with Holm-Sidak test.

absent in all beagle livers. Possible sighthound breed-associated heterogeneity was confirmed for c.943C, which was in near complete genetic linkage with c.1710G as the *POR*-H3 haplotype. *POR*-H3 was most common in greyhounds, and several other sighthound breeds, but was rare in almost all other breeds evaluated. On the other hand, c.1710G, when present without c.943C as *POR*-H2, was found in more breeds than *POR*-H3 and did not segregate between sighthound and other dog breeds. Consequently, any potential effect of *POR*-H3 on drug metabolism facilitated by POR is likely to be observed primarily in greyhounds and several other sighthound breeds, which is consistent with the known breed distribution of CYP2B11 poor metabolizers.

The *POR* c.943 G/C and c.1710 C/G cSNPs result in amino acid substitutions Glu315Gln and Asp570Glu, respectively. Multiple sequence alignments of dog POR with homologs from other species indicated that a positively charged amino acid at position 315 is highly conserved across species, suggesting its potential importance in the structure and activity of the enzyme. Conversely, the amino acid at position 570 appears to be conserved only in mammals, and both aspartate and glutamate are positively charged amino acids, suggesting that the amino acid substitution may not have a significant effect on the enzyme. PolyPhen-2 results confirmed observations from the multiple sequence alignments, predicting that Glu315Gln would be damaging, while Asp570Glu was benign. This contention was further supported by POR protein structural modelling with dynamic simulations that showed increased structural flexibility with the Glu315Gln mutation in association with loss of a salt bridge between Glu315 and Arg519. Conversely, the Asp570Glu mutation had no substantial effect on protein flexibility or atomic contacts.

These preliminary *in silico* predictions were then evaluated by functional studies of recombinant enzymes expressed in insect cells. POR protein immunoblotting showed no significant differences in the relative amounts of POR variant protein compared to POR-H1 when expressed alone or coexpressed with CYP2B11 or CYP2D15 indicating that the POR mutations do not substantially affect protein expression or degradation. Furthermore, results from the cytochrome c activity assay, revealed no substantial differences in POR enzyme kinetic

parameters between POR-H1 and each POR variant when expressed alone indicating that these POR mutations do not substantially affect intrinsic POR enzyme function. Interestingly, cytochrome c activity was significantly reduced when CYP2B11 was coexpressed with POR-H4, relative to coexpression with POR-H1. However, there was no effect on cytochrome c activity when CYP2D15 was coexpressed with POR-H4 (or any other variant). Furthermore, evaluation of P450 specific enzyme function in the same preparations showed significant reductions in CYP2B11 enzyme function (reduced Vmax values) when co-expressed with POR-H3 and POR-H4 as compared with POR-H1. However, CYP2D15 catalytic function was unaffected by coexpression with any of the POR variants. These differences in CYP2B11 function could not be explained by differences in P450 enzyme concentration since immunoblot analysis showed no differences in CYP2B11 (or CYP2D15) protein content between P450-POR preparations. Taken together, these findings indicate that the POR-H3 and POR-H4 mutations exert CYP isoform-dependent effects on enzyme function, possibly through an effect on electron transfer between POR and CYP2B11, but not between POR and CYP2D15.

Strong correlations were also observed between $V_{max}$ values for all CYP2B11 substrates tested and all CYP2D15 substrates tested. This suggests that our findings may be generalizable to other substrates of these enzymes, and the effect of POR-H3 and H4 on CYP2B11 (and lack of effect on CYP2D15) may not be substrate-dependent as has been shown for some other naturally occurring POR variants [20,24,56]. Additional studies examining other CYP2B11 and CYP2D15 substrates are warranted to confirm this.

POR is the sole redox partner of most cytochromes P450 in the endoplasmic reticulum, as well as other proteins like heme oxygenase, and therefore, needs to accommodate a range of different structures [47]. Moreover, POR exists in multiple conformation states, which may have differences in interaction and activities with different redox partners [48]. Large variability in the interactions of POR with its redox partners has been shown to result in variable effects of the same POR mutations on different P450 enzymes, including CYP2C9, CYP2C19 and CYP3A5 [49]. This mechanism might also explain the experimentally observed differences in activities of CYP2B11 and CYP2D15 when paired with different variants of canine POR, although additional work is needed to confirm this.

Human POR amino acid variants that do not result in a loss of co-factor binding can cause conformational changes that affect the interaction of POR with redox partners in a protein-dependent manner [50]. This may impact the activities of some redox partners while showing activities similar to that of wild-type POR with other partners [50]. These interactions can be based on the shape of proteins as well as atomic charge pairs, which are redox-partner-dependent and may depend on the geometry of the individual redox-partner docking sites, with the FMN domain being the preferred interaction site [50,51]. CYP2B11 co-expression with POR-H3 and -H4 caused significantly reduced $V_{max}$ values but not $K_m$ values for CYP2B11-mediated metabolism compared to co-expression with H1, indicating a disruption in the efficient transfer of electrons to CYP2B11 by POR variant Glu315Gln. It is likely that the Glu315Gln substitution affects the binding to CYP2B11 but not CYP2D15 with POR, which would also be dependent on bound substrate as substrate binding may influence POR-P450 interaction. Protein-protein interactions between the POR, CYP, and substrate may also cause conformational changes in the binding site of the interacting CYP [50,52,53], although this is less likely to be the case for the Glu315Gln substitution, given that CYP2B11 substrate $K_m$ values, indicating substrate affinity of the enzyme, were not altered and no evidence of substrate-dependent effects of enzyme kinetics for CYP2B11 were observed. Molecular dynamics simulation of the CYP2B11-POR interaction should be modeled in the future to better understand complex protein-protein interactions.

Interestingly, despite both haplotypes containing the Glu315Gln amino acid substitution, POR-H3 caused a 37% reduction in CYP2B11 $CL_{int}$ compared to the activity supported by H1, whereas H4 caused a 69% reduction in $CL_{int}$, nearly two-fold greater than H3. In human POR the presence of the POR*28 allele (A503V) is known to cause increased activities of multiple drug metabolizing cytochromes P450 and is present in different populations with a minor allele frequency ranging from 15 to 45%. The POR*28 variant in patients with POR deficiency is present together with disease-causing variants in POR and may be responsible for some of the variations in the impact of the same mutation in different patients. Analysis of vibrational entropy energy differences between wild-type and mutant structures also suggests that increased flexibility in one part of the protein due to Glu315Gln, when combined with decreased flexibility due to Asp570Glu mutation in another part of the protein may have a different impact on canine POR structure compared to either mutation alone.

While POR-H2 was not found to affect CYP2B11 or CYP2D15 activities and *in silico* analyses predicted that it would have little to no effect on enzyme stability, it cannot be ruled out that it is a completely benign mutation, given that the effects of POR variants are redox-partner-dependent. Additionally, the haplotype appeared to be localized in a select group of breeds. Doberman pinschers and rottweilers, part of the same genetic clade [54], were found to possess the highest and second highest H2 allele frequencies among all breeds screened (50% and 18.6%, respectively). Both breeds are reported to suffer from delayed sulfonamide hypersensitivity reactions more than other breeds [55]. In dogs, the reduction of the reactive sulfonamide hydroxylamine metabolite is catalyzed by cytochrome $b_5$, which accepts electrons from cytochrome $b_5$ reductase and POR. A correlation between a cytochrome $b_5$ reductase polymorphism and sulfonamide hypersensitivity exists, but hypersensitivity cannot be explained by the polymorphism alone [56,57]. Electron transfer between POR variants and cytochrome $b_5$ should be examined in the future.

As with the *CYP2B11* variant haplotypes, POR haplotype frequencies differences between AKC-registered greyhound and NGA-registered greyhounds were observed [25]. Most notably, the NGA greyhounds had an H3 allele frequency that was more than three-fold greater than AKC greyhounds. These population differences in haplotype frequency could be explained by a founder effect at the initial separation of the two populations or it may be a consequence of selective breeding for the different phenotypes that characterize the NGA versus AKC breeds.

The results of this study predict that some, but not all, greyhounds and related sighthound breeds, such as the Scottish deerhound, would have reduced CYP2B11-mediated metabolism resulting from these *POR* variants. Activity is expected to be reduced by approximately 37% in dogs with the *POR*-H3/H3 diplotype compared to dogs with the *POR*-H1/H1 diplotype. Dogs with the rare *POR* H4/H4 diplotype are expected to have an activity reduction of 69%. For human POR variant functional studies, in general, *in vitro* assay results tend to correlate well with *in vivo* clinical observations [20,58–61]. However, given that this is the first report of canine POR variants, it is unknown whether the *in vitro* assay results described in this study will accurately predict effects on drug pharmacokinetics *in vivo*.

There are some limitations to the current study. POR-H1 and POR viral coinfections were optimized to result in maximal P450 enzyme activities. We did not measure the absolute concentration of expressed POR protein in the experimental samples and so it unclear whether the content of POR relative to P450 protein is similar to that found *in vivo*. Furthermore, although our *in vitro* data indicated that the POR mutations did not affect expressed POR protein content, measurement of the POR protein content in liver tissue from dogs with different POR genotypes would be needed to confirm this. Our predictions regarding the impact of *POR* variant haplotypes on CYP2B11 and CYP2D15 activities are solely based on *in vitro* studies with

extrapolation *in vivo*. The combined effects of *CYP2B11* and *POR* haplotype variants on CYP2B11 function *in vivo* also remain unknown. Accordingly, future studies are necessary to confirm and expand our findings by evaluating CYP2B11, CYP2D15, and other drug-metabolizing CYP isoform functions *in vivo* using isoform-specific drug phenotyping probes comparing dogs from different breeds and with different *POR* and *CYP2B11* genotypes.

## Supporting information

**S1 Fig. *POR* haplotype frequencies in 68 dog breeds and mixed breed dogs.**
(PDF)

**S2 Fig. Full-length immunoblots of cropped blots displayed in Fig 5A–5C.**
(PDF)

**S3 Fig. Enzyme activity correlation plots for CYP2B11 coexpressed with POR variants.**
(PDF)

**S4 Fig. Enzyme activity correlation plots for CYP2D15 coexpressed with POR variants.**
(PDF)

**S1 Table. *POR* gene PCR and sequencing primers.**
(PDF)

**S2 Table. Primers and reporters used for *POR* genotyping.**
(PDF)

**S3 Table. Genetic polymorphisms identified in the coding region of the POR gene in 5 beagles and 13 greyhounds.**
(PDF)

**S4 Table. POR genotype frequencies in 68 dog breeds and mixed breed dogs.**
(PDF)

**S1 File. Raw data underlying all mean and standard deviation values presented in the manuscript.**
(PDF)

**S1 Data. Raw data underlying all mean and standard deviation values presented in the manuscript.**
(DOCX)

## Acknowledgments

The authors thank Dr. Katrina L. Mealey, Rebecca L. Connors, and the Veterinary Teaching Hospital DNA Bank at Washington State University for the contribution of DNA samples that were used for genotyping.

## Author Contributions

**Conceptualization:** Stephanie E. Martinez, Michael H. Court.

**Data curation:** Stephanie E. Martinez, Michael H. Court.

**Formal analysis:** Amit V. Pandey, Michael H. Court.

**Funding acquisition:** Michael H. Court.

**Investigation:** Stephanie E. Martinez, Amit V. Pandey, Zhaohui Zhu, Michael H. Court.

**Methodology:** Stephanie E. Martinez, Tania E. Perez Jimenez, Michael H. Court.

**Resources:** Stephanie E. Martinez, Zhaohui Zhu, Michael H. Court.

**Validation:** Tania E. Perez Jimenez.

**Writing – original draft:** Stephanie E. Martinez, Amit V. Pandey.

**Writing – review & editing:** Stephanie E. Martinez, Amit V. Pandey, Tania E. Perez Jimenez, Michael H. Court.

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
