## [Decision Letter · Decision Letter 0]

18 Sep 2023

PONE-D-23-24955Pharmacogenomics of poor drug metabolism in greyhounds: Canine P450 oxidoreductase genetic variation, breed heterogeneity, and functional characterizationPLOS ONE

Dear Dr. Court,

Thank you for submitting your manuscript to PLOS ONE. After careful consideration, we feel that it has merit but does not fully meet PLOS ONE’s publication criteria as it currently stands. Therefore, we invite you to submit a revised version of the manuscript that addresses the points raised during the review process. Please pay close attention to the comments from the reviewers when revising your manuscript and carefully address each point raised by the reviewers.

We look forward to receiving your revised manuscript.

Kind regards,

Jed N. Lampe, Ph.D.

Academic Editor

PLOS ONE

Journal Requirements:

5. Please ensure that you refer to Figure 9 in your text as, if accepted, production will need this reference to link the reader to the figure.

Additional Editor Comments:

Please carefully address all comments provided by the reviewers, particularly those provided by reviewer number two. Also please ensure that your revised manuscript conforms to all PLOS ONE standards and guidelines, including those regarding figures and tables.

Reviewers' comments:

Reviewer's Responses to Questions

**Comments to the Author**

1. Is the manuscript technically sound, and do the data support the conclusions?

Reviewer #1: Partly

Reviewer #2: Partly

2. Has the statistical analysis been performed appropriately and rigorously? 

Reviewer #1: Yes

Reviewer #2: Yes

3. Have the authors made all data underlying the findings in their manuscript fully available?

Reviewer #1: Yes

Reviewer #2: No

4. Is the manuscript presented in an intelligible fashion and written in standard English?

Reviewer #1: Yes

Reviewer #2: Yes

5. Review Comments to the Author

Reviewer #1: Stephanie and her colleagues conducted a study aimed at identifying POR gene mutants that could help to elucidate the diminished CYP2B11-mediated metabolism observed in greyhounds. In pursuit of this goal, comprehensive gene analysis was executed, leading to the discovery of two highly specific single nucleotide polymorphisms (SNPs) in greyhounds. These SNPs delineated four distinct haplotypes, with one haplotype showing notably higher prevalence among greyhounds. To delve deeper into the influence of various protein mutants on CYP450, four variants were expressed in insects for further investigation. Subsequent functional analyses revealed the impact of mutant H3 (which is predicted to be more prevalent in greyhounds) on the reduction of CYP2B11 activity. This finding provides valuable insight into the diminished CYP2B11-mediated metabolism in greyhounds. Overall, the manuscript is well-written, offering clear explanations and comprehensive detailing of the experimental methodologies employed.

However, one major concern I have regarding this manuscript is that gene-level observations do not inherently translate to protein expression. While the prevalence of H3 mutation in greyhounds at the gene level is notable, it does not automatically imply that the H3 protein mutant will be expressed within liver tissue. The authors lack data illustrating protein expression levels for both total POR and various mutants in liver tissue across different dog breeds. Evidence on protein level in liver tissue should be included.

Additionally, there are a few minor points that require attention:

1) In this manuscript, the corresponding author is Michael; however, a discrepancy exists wherein on both page 1 of the main manuscript and the supplementary information, the asterisk denoting the corresponding author appears after Stephanie's name. Moreover, the supplementary information erroneously retains Stephanie's email address as the contact information for the corresponding author.

2) It is advised to provide a succinct but comprehensive description of the methodologies employed for allele frequency calculation, haplotype frequency calculation, and CLint calculation.

3) In line 136-138, the authors state “As shown in Table S1, all 13 greyhound DNA samples tested had one (5 of 13) or both (8 of 13) of these nonsynonymous SNPs.” However, Table S1 reflects that among the 13 DNA samples, one SNP was present in 4 out of 13 samples, and both SNPs were present in 9 out of 13 samples.

4) In line 185-187, the authors mention “We combined the best parts of the 5 models to obtain a hybrid model, to increase the accuracy beyond each of the contributors.” Can the authors further elaborate on what do they mean by the best parts of the 5 models?

5) In line 234-236, the authors state “However the mean (± SD) NADPH Km value for H4 (2.5 ± 0.3 μM) was significantly lower than the Km for H1 (13.2 ± 0.2 μM) (P = 0.045, ANOVA with Holm-Sidak test).” However, a discrepancy arises as Table 2 indicates a Km value of 3.2 ± 0.2 μM for H1. The authors are advised to verify and rectify the Km values for accuracy.

6) In line 266-269, the authors state “Furthermore, Vmax values for 7-benzyloxyresorufin O-debenzylation, propofol hydroxylase, and bupropion hydroxylase for each POR variant were also highly correlated with cytochrome c (POR) activity (Rs ≥ 0.74, P < 0.001) as shown in Fig. S3D-F.” However, a discrepancy arises as Figure S3F indicates the Rs value of 0.735, and P value of 0.00805. Additionally, the colors indicating POR-H1 and POR-H2 are inconsistent with the legend in Figure S3F.

7) In line 362, there is a typo “substrate-dependen”.

8) The format of the reference needs to be double checked, some references have the doi, while some do not. Consistency in the reference format needs to be ensured.

Reviewer #2: In this manuscript Martinez and co-authors describe the investigation of the genetic variation of canine P450 oxidoreductase in their pursuit in explaining the discrepancy of canine CYP2B11 metabolism in greyhounds (and some other sighthound breeds) versus other dog breeds. This discrepancy was noted, as described in a former study of this group (ref 4), when analyzing the genetic variation of canine CYP2B11. Authors describe two mutations (Glu315Gln and Asp570Glu) in POR of greyhounds, of which particular the former one is indicated to be responsible for the P450 isoform specific slow metabolizer phenotype, observed in greyhounds.

Authors used several methodologies which seem in fact to indicate that the Glu315Gln mutation (haplotype 4) is responsible for decreased activity of CYP2B11, which is observed to a lesser extent with the double mutant (haplotype H3). This seems not to be the case for dog CYP2D15 for both H3 and H4. Authors present in silico/modelling data, rationalizing that mutations cause this P450 isoform effect, however presented data hardly underpin this conclusion. Some doubts remain on additional issues which should be addressed.

Major issues.

1. Although authors present a set of in silico and modelling data, this hardly allows the conclusion (lines 199-201) or prediction (lines 321-322) for a P450 isoform effect. Based on presented data, one may indicate that the Glu315Gln mutation seem to cause an overall stability change of the protein, although no experimental data is presented to verify this effect (e.g., CD spectroscopy). No specific data is presented that this mutation may interfere with two main mechanisms, currently hold responsible for the P450 isoform specific effects, namely the open/closed dynamics of the POR protein, and the FMN domain, the interaction site of P450s for electron transfer. Although authors recognize the importance of the extensive protein dynamics of POR in its electron donation function (lines 325-326), and variability of interaction of redox partners with POR (lines 326-329), based on presented data one can only speculate on how the Glu315Gln mutation could interfere with the dynamics and the FMN interaction domain. Furthermore, only one other canine P450 (2D15) was used, which calls for caution in drawing conclusions (lines 354-355) regarding the isoform effect on CYP2B11. This issue should be corrected both in the Result and Discussion section.

2. Lines 210-222 - expression levels of POR variants and P450s: doubts remain regarding the quantification of POR variants and P450s in Sf9 microsomes. Figure 5 demonstrates quite some differences in intensity of POR bands (Figures 5B: H1 versus H2, 3 and 4, and Fig 5C: H4 versus H1, 2 and 3) and difference in quality of bands intensity of POR between Figure 5 and Figure S2. No explanation is given for the smaller CYP2D15 bands in Figure S2. Line 558: incorrect use of this extinction coefficient; Sf9 insect cells do not contain (or very little) hemoglobin’s, so the extinction coefficient of 91,000 M-1 cm-1 should have been used (see ref 58). Only relative expression levels are presented, no absolute protein contents of POR variants and of the two used P450s in insect microsomes are presented; such information (Table) should be added to Supplementary Information.

3. Lines 243-247: these results are in unexpected and contradictory, taking the results described in lines 216-219 into account, no interpretation is given, which is quite pertinent.

Minor issues

1) Introduction and Discussion sections are very extensive, could be improved.

2) Line 119: rational for usage of canine CYP2D15 should be given here and not in the Discussion section (line 353)

3) Line 180: the usage of only one web-based tool to explore the effects of mutations on the structure and function of proteins is very limited (see e.g., doi: 10.1371/journal.pone.0267084); the additional use of one or two additional platforms (e.g., PROVEAN, SIFT, SNPs&GO or PhD-SNP) is advisable.

4) Lines 195-199: kCal should be kcal

5) Lines 332-324: incorrect: recombinant expression systems in E.coli for full length POR (with or without co-expression of P450s) exist.

6) Lines 336-338: what POR:P450 ratio is considered optimal, maximum P450 activity or ratios reflecting in vivo stoichiometry’s? What was the POR:CYP ratio used in this study? (See point 2, Major issues)

7) Line 362: “independent”

8) Line 364: write: “… by some natural occurring POR variants…”. Several POR mutations have shown to cause P450 isoform and substrate dependent effects, see: doi:10.3390/ijms21186669 and included references.

9) Line 392: all microsomal P450s dock on the FMN domain of POR for electron transfer.

10) Line 395: not necessarily, this may depend on the substrate bound.

11) Lines 409-415: correct, however data presented seem to indicate a compensatory effect of the Asp570Glu mutation on the effect of Glu315Gln, at least for the substrates tested. It would be interesting to see the effect of both mutations simultaneously on POR’s flexibility (Fig 4).

12) Lines 422-428: sulfonamide metabolism is not restricted to phase I metabolism. Sulfonamides are metabolized hepatically by oxidation, acetylation, and/or glucuronidation; genetic polymorphism of conjugation (phase II) enzymes (frequently occurring) may play an additional important role in observed sulfonamide susceptibility.

13) Lines 545-546: which were the final used MOIs, determined in preliminary experiments?

14) Lines 621, 630 and 630: give P450 concentrations, as are given for the CYP2D15 assays.

15) Lines 619-645: solvents may strongly influence P450 activity (e.g., see PMID: 9929510); which solvents and final concentrations were used? Were solvent concentrations kept constant along the tested substrate concentration range?

16) Tables 1-4 have very long descriptions (several times duplicated from the Materials and Method section), should be downsized, and placed as footnotes.

17) Supplementary Information: Figure S2 correct to “Figure 5A, 5B and 5C.

6. PLOS authors have the option to publish the peer review history of their article (what does this mean?). If published, this will include your full peer review and any attached files.

Reviewer #1: No

Reviewer #2: No

---

## [Author Response · Author response to Decision Letter 0]

13 Dec 2023

Dear Editor,

Thank you for obtaining the careful critique of this manuscript. Apologies for the delay in response that was a consequence of some communication problems with one of the coauthors that we have solved recently. The following are our responses to your requests. Responses to the reviewers are provided separately.

EDITOR: 

1. Please ensure that your manuscript meets PLOS ONE's style requirements. 

The manuscript has been edited to meet these requirements.

2. Please note that funding information should not appear in any section or other areas of your manuscript.

Funding information has been removed from the text.

3. In your Data Availability statement, you have not specified where the minimal data set underlying the results described in your manuscript can be found.

Computer model data used for the POR dynamic structural modelling are available here https://doi.org/10.48620/379 . All other underlying data, including blot images, used to generate the graphs and tables shown in the manuscript are provided in the Supporting Information (S1 File).

4. PLOS ONE now requires that authors provide the original uncropped and unadjusted images underlying all blot or gel results reported in a submission’s figures or Supporting Information files.

All blots are now provided in the Supporting Information (S1 File). 

5. Please ensure that you refer to Figure 9 in your text as, if accepted, production will need this reference to link the reader to the figure.

Figure 9 reference has been added.

6. Please include captions for your Supporting Information files at the end of your manuscript, and update any in-text citations to match accordingly.

Supporting Information files are now captioned at the end and the appropriate citations are included in the text. 

Dear Reviewers

Thank you for your careful critique of this manuscript. Apologies for the delay in response that was a consequence of some communication problems with one of the coauthors that we have solved recently. The following are our responses to the questions, comments, and requests that were raised. Changes in the manuscript are highlighted in the marked-up document with references to the Line numbers below.

REVIEWER #1: Stephanie and her colleagues conducted a study aimed at identifying POR gene mutants that could help to elucidate the diminished CYP2B11-mediated metabolism observed in greyhounds. In pursuit of this goal, comprehensive gene analysis was executed, leading to the discovery of two highly specific single nucleotide polymorphisms (SNPs) in greyhounds. These SNPs delineated four distinct haplotypes, with one haplotype showing notably higher prevalence among greyhounds. To delve deeper into the influence of various protein mutants on CYP450, four variants were expressed in insects for further investigation. Subsequent functional analyses revealed the impact of mutant H3 (which is predicted to be more prevalent in greyhounds) on the reduction of CYP2B11 activity. This finding provides valuable insight into the diminished CYP2B11-mediated metabolism in greyhounds. Overall, the manuscript is well-written, offering clear explanations and comprehensive detailing of the experimental methodologies employed.

REVIEWER #1: However, one major concern I have regarding this manuscript is that gene-level observations do not inherently translate to protein expression. While the prevalence of H3 mutation in greyhounds at the gene level is notable, it does not automatically imply that the H3 protein mutant will be expressed within liver tissue. The authors lack data illustrating protein expression levels for both total POR and various mutants in liver tissue across different dog breeds. Evidence on protein level in liver tissue should be included.

RESPONSE: Agreed. We have previously quantified POR protein in livers from 59 different dogs (DOI: 10.1124/dmd.119.088070), including 5 greyhounds (DL03, DL04, DL08, DL09, DL10). The genotypes for these livers are given in S3 Table. All 5 greyhounds had at least one POR mutation (H2 or H3), and 2 of these were homozygous H3/H3. The POR content for these two H3/H3 livers were 32 and 30 pmoles/mg microsomes, while the POR content of all 5 greyhound livers ranged from 24 to 42 pmoles/mg. For all 59 livers tested, the median (5% to 95%) POR content was 40 (24 to 58) pmoles/mg, and the range was 19 to 85 pmoles/mg. These data suggest that liver POR content in greyhounds, including those with the POR-H3/H3 genotype, do not differ substantially from non-greyhound livers. However, these are too small numbers of livers resulting in insufficient power for statistical hypothesis testing. Consequently we did not include these data in the final manuscript given the requirement of PLOS One for robust data Furthermore we do not have access to additional liver tissues with these specific mutations across different breeds that could be tested. Obtaining such liver samples from different breeds is quite challenging since dog owners are reluctant to donate tissues from their pets. It is more straightforward to obtain liver samples from purpose-bred research dogs. However these dogs are primarily beagles, which do not have these POR mutations, based on testing 50 different beagle dogs in this study (S1 Figure). Furthermore, none of the 25 beagle dog livers in our liver bank had any POR mutation. In any case, we have added this as a limitation of the study in the discussion as follows: “Furthermore, although our in vitro data indicated that the POR mutations did not affect expressed POR protein content, measurement of the POR protein content in liver tissue from dogs with different POR genotypes would be needed to confirm this.” Lines 750-753.

REVIEWER #1: Additionally, there are a few minor points that require attention:

1) In this manuscript, the corresponding author is Michael; however, a discrepancy exists wherein on both page 1 of the main manuscript and the supplementary information, the asterisk denoting the corresponding author appears after Stephanie's name. Moreover, the supplementary information erroneously retains Stephanie's email address as the contact information for the corresponding author.

RESPONSE: Corrected.

REVIEWER #1: 2) It is advised to provide a succinct but comprehensive description of the methodologies employed for allele frequency calculation, haplotype frequency calculation, and CLint calculation.

RESPONSE: Methods used to calculate genotype and haplotype (allele) frequencies are now given at lines 176-178. The method to calculate CLint is now given at lines 299-300.

REVIEWER #1: 3) In line 136-138, the authors state “As shown in Table S1, all 13 greyhound DNA samples tested had one (5 of 13) or both (8 of 13) of these nonsynonymous SNPs.” However, Table S1 reflects that among the 13 DNA samples, one SNP was present in 4 out of 13 samples, and both SNPs were present in 9 out of 13 samples.

RESPONSE: Thank you for identifying that error. Indeed 2 greyhounds had the reference genotype. This statement has been corrected. It now reads “As shown in S3 Table, 11 of 13 greyhound DNA samples tested had one (2 of 11) or both (9 of 11) of these nonsynonymous SNPs.” Lines 349-351

REVIEWER #1: 4) In line 185-187, the authors mention “We combined the best parts of the 5 models to obtain a hybrid model, to increase the accuracy beyond each of the contributors.” Can the authors further elaborate on what do they mean by the best parts of the 5 models?

RESPONSE: When a hybrid model is built, “bad” (poorest scoring) regions in the top scoring model are iteratively replaced with corresponding fragments from the other models. The model used to seed this hybridization is chosen using a combined score that considers model quality, alignment accuracy, sequence similarity and match with predicted secondary structure. By using this process, the best scoring parts of each of the individual models are chosen to put together a final model that usually improves on the individual models based on coverage of regions in the PDB files used for model building. An advantage is also the greater coverage of residues taken from multiple PDB files. The text has been edited to clarify this (Line 193; Lines 409-412)

REVIEWER #1: 5) In line 234-236, the authors state “However the mean (± SD) NADPH Km value for H4 (2.5 ± 0.3 μM) was significantly lower than the Km for H1 (13.2 ± 0.2 μM) (P = 0.045, ANOVA with Holm-Sidak test).” However, a discrepancy arises as Table 2 indicates a Km value of 3.2 ± 0.2 μM for H1. The authors are advised to verify and rectify the Km values for accuracy.

RESPONSE: Thank you for identifying this error. This sentence has been corrected to read: “However the mean (± SD) NADPH Km value for H4 (2.5 ± 0.3 µM) was significantly lower than the Km for H1 (3.2 ± 0.2 µM) (P = 0.045, ANOVA with Holm-Sidak test).” (lines 498-500)

REVIEWER #1: 6) In line 266-269, the authors state “Furthermore, Vmax values for 7-benzyloxyresorufin O-debenzylation, propofol hydroxylase, and bupropion hydroxylase for each POR variant were also highly correlated with cytochrome c (POR) activity (Rs ≥ 0.74, P < 0.001) as shown in Fig. S3D-F.” However, a discrepancy arises as Figure S3F indicates the Rs value of 0.735, and P value of 0.00805. Additionally, the colors indicating POR-H1 and POR-H2 are inconsistent with the legend in Figure S3F.

RESPONSE: Thank you for identifying these errors. The sentence now reads “Furthermore, Vmax values for 7-benzyloxyresorufin O-debenzylation, propofol hydroxylase, and bupropion hydroxylase for each POR variant were also highly correlated with cytochrome c (POR) activity (Rs ≥ 0.73, P < 0.001) (S3D-F Fig).” Lines 579-582 The colors in S3F Fig also have been corrected to match the legend.

REVIEWER #1: 7) In line 362, there is a typo “substrate-dependen”.

RESPONSE: Corrected. Line 677

REVIEWER #1: 8) The format of the reference needs to be double checked, some references have the doi, while some do not. Consistency in the reference format needs to be ensured.

RESPONSE: The reference style for PloS One requests inclusion of the DIGITAL OBJECT IDENTIFIER (DOI) if available as part of the reference. However, not all references have a DOI. I have reviewed and added DOIs if they can be identified.

REVIEWER #2: In this manuscript Martinez and co-authors describe the investigation of the genetic variation of canine P450 oxidoreductase in their pursuit in explaining the discrepancy of canine CYP2B11 metabolism in greyhounds (and some other sighthound breeds) versus other dog breeds. This discrepancy was noted, as described in a former study of this group (ref 4), when analyzing the genetic variation of canine CYP2B11. Authors describe two mutations (Glu315Gln and Asp570Glu) in POR of greyhounds, of which particular the former one is indicated to be responsible for the P450 isoform specific slow metabolizer phenotype, observed in greyhounds.

Authors used several methodologies which seem in fact to indicate that the Glu315Gln mutation (haplotype 4) is responsible for decreased activity of CYP2B11, which is observed to a lesser extent with the double mutant (haplotype H3). This seems not to be the case for dog CYP2D15 for both H3 and H4. Authors present in silico/modelling data, rationalizing that mutations cause this P450 isoform effect, however presented data hardly underpin this conclusion. Some doubts remain on additional issues which should be addressed.

Major issues.

REVIEWER #2: 1. Although authors present a set of in silico and modelling data, this hardly allows the conclusion (lines 199-201) or prediction (lines 321-322) for a P450 isoform effect. Based on presented data, one may indicate that the Glu315Gln mutation seem to cause an overall stability change of the protein, although no experimental data is presented to verify this effect (e.g., CD spectroscopy). No specific data is presented that this mutation may interfere with two main mechanisms, currently hold responsible for the P450 isoform specific effects, namely the open/closed dynamics of the POR protein, and the FMN domain, the interaction site of P450s for electron transfer. Although authors recognize the importance of the extensive protein dynamics of POR in its electron donation function (lines 325-326), and variability of interaction of redox partners with POR (lines 326-329), based on presented data one can only speculate on how the Glu315Gln mutation could interfere with the dynamics and the FMN interaction domain. Furthermore, only one other canine P450 (2D15) was used, which calls for caution in drawing conclusions (lines 354-355) regarding the isoform effect on CYP2B11. This issue should be corrected both in the Result and Discussion section.

RESPONSE: We agree. We have edited the statement in the results and discussion as follows: 

Original lines 199-201 from the results now reads: “Disruption to the structural stability of the enzyme could potentially lead to a change in interaction with cytochrome P450 partner proteins and impact the activities of cytochromes P450 that require POR as their redox partner.” Lines 449-451.

Original lines 318-325 from the discussion: These were deleted and a shorter discussion of the protein flexibility result is provided here: “ This contention was further supported by POR protein structural modelling with dynamic simulations that showed increased structural flexibility with the Glu315Gln mutation in association with loss of a salt bridge between Glu315 and Arg519. Conversely, the Asp570Glu mutation had no substantial effect on protein flexibity or atomic contacts.” Lines 648-652.

Original lines 325-329 from the discussion have been replaced with “POR is the sole redox partner of most cytochromes P450 in the endoplasmic reticulum, as well as other proteins like heme oxygenase, and therefore, needs to accommodate a range of different structures [47]. Moreover, POR exists in multiple conformation states, which may have differences in interaction and activities with different redox partners [48]. Large variability in the interactions of POR with its redox partners has been shown to result in variable effects of the same POR mutations on different P450 enzymes, including CYP2C9, CYP2C19 and CYP3A5 [49]. This mechanism might also explain the experimentally observed differences in activities of CYP2B11 and CYP2D15 when paired with different variants of canine POR, although additional work is needed to confirm this.” Lines 679-687.

Original lines 354-355 from the discussion: Most of this paragraph (from Lines 349 to 358) was deleted to reduce discussion length as requested. The remaining part of the paragraph concerning substrate dependency has been edited as follows: “Strong correlations were also observed between Vmax values for all CYP2B11 substrates tested and all CYP2D15 substrates tested. This suggests that our findings may be generalizable to other substrates of these enzymes, and the effect of POR-H3 and H4 on CYP2B11 (and lack of effect on CYP2D15) may not be substrate-dependent as has been shown for some other naturally occurring POR variants [20,24,56]. Additional studies examining other CYP2B11 and CYP2D15 substrates are warranted to confirm this.” Lines 673-678.

REVIEWER #2: 2. Lines 210-222 - expression levels of POR variants and P450s: doubts remain regarding the quantification of POR variants and P450s in Sf9 microsomes. Figure 5 demonstrates quite some differences in intensity of POR bands (Figures 5B: H1 versus H2, 3 and 4, and Fig 5C: H4 versus H1, 2 and 3) and difference in quality of bands intensity of POR between Figure 5 and Figure S2. No explanation is given for the smaller CYP2D15 bands in Figure S2. Line 558: incorrect use of this extinction coefficient; Sf9 insect cells do not contain (or very little) hemoglobin’s, so the extinction coefficient of 91,000 M-1 cm-1 should have been used (see ref 58). Only relative expression levels are presented, no absolute protein contents of POR variants and of the two used P450s in insect microsomes are presented; such information (Table) should be added to Supplementary Information.

RESPONSE: Regarding “differences in intensity of POR bands (Figures 5B: H1 versus H2, 3 and 4, and Fig 5C: H4 versus H1, 2 and 3)” – The blots shown in A, B, and C. are examples of only one blot (each) from 3 replicate blots of each of 4 protein preps (12 total blots). So differences are to be anticipated between replicates and between protein preparations that are then averaged. Differences between protein preparations are reflected by the standard deviation bars shown below the blots. We have provided all blots and derived data in the Supplementary Information file – S1 File. 

Regarding “difference in quality of bands intensity of POR between Figure 5 and Figure S2.” We have copied the cropped blots directly from S2 Fig to Fig. 5 to ensure they have identical quality. 

Regarding “No explanation is given for the smaller CYP2D15 bands in Figure S2. “ This was an oversight. In the legends to S2 Fig and S7 Fig (contained in S1 File) we have now pointed out this second band, that migrates more rapidly (smaller size) than the CYP2D15 band. Since it is present in all Sf9 microsome samples (including cells not infected with CYP2D15 virus), but is not found in dog liver microsomes, it appears to be a crossreactive protein expressed in Sf9 cells. 

Regarding “Line 558: incorrect use of this extinction coefficient; Sf9 insect cells do not contain (or very little) hemoglobin’s, so the extinction coefficient of 91,000 M-1 cm-1 should have been used (see ref 58).” This was an error resulting from a misinterpretation of ref 58. We have corrected all data (graphs and tables) that were normalized to P450 concentration.

Regarding: “Only relative expression levels are presented, no absolute protein contents of POR variants and of the two used P450s in insect microsomes are presented; such information (Table) should be added to Supplementary Information.” We did not have pure canine POR standard that would allow us to determine the absolute protein content of the POR variants. The main goal was to determine whether there were POR protein differences between POR-H1 and the POR variants despite infecting the same amount of POR expressing virus. The POR and P450 blots and associated quantitative data are now provided in the Supplementary Information S1 File. 

REVIEWER #2: 3. Lines 243-247: these results are in unexpected and contradictory, taking the results described in lines 216-219 into account, no interpretation is given, which is quite pertinent.

RESPONSE: Agreed. This was an oversight and is now discussed in paragraph 3 of the discussion. Lines 653-672

Minor issues

REVIEWER #2: 1) Introduction and Discussion sections are very extensive, could be improved.

RESPONSE: We have substantially reduced the length of the Discussion (about 1 page shorter) by elimination of some redundancy and sections that do not directly address the results. We were unable to substantially shorten the introduction and at 2 ½ pages we believe this is the minimum to justify the reported study.

REVIEWER #2: 2) Line 119: rational for usage of canine CYP2D15 should be given here and not in the Discussion section (line 353)

RESPONSE: This sentence has been moved from the Discussion to the Introduction Lines 117-118. 

REVIEWER #2: 3) Line 180: the usage of only one web-based tool to explore the effects of mutations on the structure and function of proteins is very limited (see e.g., doi: 10.1371/journal.pone.0267084); the additional use of one or two additional platforms (e.g., PROVEAN, SIFT, SNPs&GO or PhD-SNP) is advisable.

RESPONSE: Thank you for this advice. We did investigate the use of the suggested tools and others reviewed in doi: 10.1371/journal.pone.0267084 . However most of the mentioned tools were either not currently accessible (PROVEAN is “retired”; SIFT web page was unresponsive) or would not evaluate dog protein sequences since input was restricted to human sequences (SNPs&GO) and/or required structural sequence data from the PDB database with resolved structure. The remaining tools were limited in accuracy of prediction (such as PhD-SNP). PolyPhen2 (which was used here) is currently considered the “gold standard” for initial screening of novel mutations, but has known limitations in sensitivity and specificity. Importantly, functional characterization of mutations, as was done in this paper, would still indicated, whether or not additional programs did (or did not) agreed with the Polphen2 finding. We do not believe that adding any other (potentially less accurate) analysis will change any of the conclusions of this paper. 

REVIEWER #2: 4) Lines 195-199: kCal should be kcal.

RESPONSE: All instances were corrected.

REVIEWER #2: 5) Lines 332-324: incorrect: recombinant expression systems in E.coli for full length POR (with or without co-expression of P450s) exist.

RESPONSE: Agreed. This statement was actually pertaining the need to N-terminally modify CYP genes (not POR). However, the main reason was that E Coli lack eukaryotic post-translational modifications present in Sf9 cells. Regardless, this paragraph is not essential and we have elected to delete to decrease the length of the Discussion – as requested above.

REVIEWER #2: 6) Lines 336-338: what POR:P450 ratio is considered optimal, maximum P450 activity or ratios reflecting in vivo stoichiometry’s? What was the POR:CYP ratio used in this study? (See point 2, Major issues)

RESPONSE: This paragraph was deleted to reduce discussion length as requested above. 

Regarding “what POR:P450 ratio is considered optimal, maximum P450 activity or ratios reflecting in vivo stoichiometry’s?” We have modifed the methods description to indicate that “optimum” means maximal POR-H1 and P450 activity for the POR:P450 coexpression experiments. Lines 216-226 

Regarding “What was the POR:CYP ratio used in this study? “ We did not measure absolute concentrations of POR, in large part because we did not have a purified canine POR protein preparation to use as a standard. It is possible that the POR:CYP ratio we used differed from the in vivo ratio. This is a limitation that has been added to the Discussion – lines 746-749. 

REVIEWER #2: 7) Line 362: “independent”

RESPONSE: Corrected. Line 676.

REVIEWER #2: 8) Line 364: write: “… by some natural occurring POR variants…”. Several POR mutations have shown to cause P450 isoform and substrate dependent effects, see: doi:10.3390/ijms21186669 and included references.

RESPONSE: Agreed. We have reworded to indicate this and added this reference. Line 675-678

REVIEWER #2: 9) Line 392: all microsomal P450s dock on the FMN domain of POR for electron transfer.

RESPONSE: Generally yes, but interactions with other domains are also likely, perhaps in a two step process where substrate binding can influence the final mode of interaction. However, the details of POR-P450 interaction need further investigation and multiple different P450-POR interaction modes need to be studied before a final conclusion can be drawn.

We have amended the sentence to indicate that FMN domain is the preferred interaction site as follows: “These interactions can be based on the shape of proteins as well as atomic charge pairs, which are redox-partner-dependent and may depend on the geometry of the individual redox-partner docking sites, with the FMN domain being the preferred interaction site” Lines 691-694

REVIEWER #2: 10) Line 395: not necessarily, this may depend on the substrate bound.

RESPONSE: Agreed. We have amended the sentence to mention substrate binding influence of POR-P450 interaction as follows: “It is likely that the Glu315Gln substitution affects the binding to CYP2B11 but not CYP2D15 with POR, which would also be dependent on bound substrate as substrate binding may influence POR-P450 interaction.” Lines 697-699

REVIEWER #2: 11) Lines 409-415: correct, however data presented seem to indicate a compensatory effect of the Asp570Glu mutation on the effect of Glu315Gln, at least for the substrates tested. It would be interesting to see the effect of both mutations simultaneously on POR’s flexibility (Fig 4).

RESPONSE: This is an interesting possibility. We did not perform these calculations to keep the analysis focused on individual effects that could be compared with WT protein, as software limitations may not give a accurate value for multiple mutations and would require remodeling the whole protein based on a different sequence that incorporates both mutations, and that would generate a new model. We will consider solving this problem in future studies.

REVIEWER #2: 12) Lines 422-428: sulfonamide metabolism is not restricted to phase I metabolism. Sulfonamides are metabolized hepatically by oxidation, acetylation, and/or glucuronidation; genetic polymorphism of conjugation (phase II) enzymes (frequently occurring) may play an additional important role in observed sulfonamide susceptibility.

RESPONSE: This is true. And we are not excluding those other pathways. However, the quoted studies (and others referred therein) attribute the hypersensitivity in these dogs to reduced detoxification of the reactive hydroxylamine metabolite of sulfonamides by cytochrome b5. Furthermore, we wanted to maintain the focus here on the potential for POR polymorphisms – in addition to the known cytochrome b5 reductase polymorphism - to influence toxicity susceptibility. 

REVIEWER #2: 13) Lines 545-546: which were the final used MOIs, determined in preliminary experiments?

RESPONSE: Absolute MOIs were not determined using a plaque assay. Rather stock viral titers were measured relative to POR-H1 using a validated QPCR assay that measures viral specific Gp64 DNA concentrations, which are directly correlated with plaque assay MOI values. Preliminary experiments then determined POR-H1 titers that resulted in maximal POR, CYP2B11 and CYP2D15 activity (respectively). The same titer of POR-H2, H3 and H4 were then used for subsequent infections. The main goal was to ensure that we infected the same amount of virus for each POR variant. 

This section has been rewritten to clarify as follows: “Amplified viral stocks were titered relative to the recombinant POR-H1 baculovirus stock using a TaqMan® gene expression assay (Thermo Fisher Scientific) that measures viral Gp64 DNA concentration by QPCR as described by Hitchman et al [38]. This assay has been validated as an accurate and reproducible alternative to plaque assay titration. Preliminary experiments were conducted to determine the POR-H1 viral titer that resulted in maximal cytochrome c reductase activity in infected cells. The same viral titer was then used in subsequent experiments to infect cells with POR-H1, each POR variant or the GUS baculoviruses. CYP2B11 and CYP2D15 viral titers that resulted in maximal 7-benzyloxyresorufin O-debenzylation and tramadol O-demethylation (respectively) when coexpressed with POR-H1 were also determined and used for coexpression experiments with POR-H1, each POR variant and PFEV control.” Lines 216-226.

REVIEWER #2: 14) Lines 621, 630 and 630: give P450 concentrations, as are given for the CYP2D15 assays.

RESPONSE: Changed to molar concentration for CYP2B11. Lines 305, 313, 316.

REVIEWER #2: 15) Lines 619-645: solvents may strongly influence P450 activity (e.g., see PMID: 9929510); which solvents and final concentrations were used? Were solvent concentrations kept constant along the tested substrate concentration range?

RESPONSE: Acetonitrile was used to dissolve substrates with final solvent concentrations less than or equal to 0.5%. Solvent concentrations were constant across the tested range of substrate concentrations. This information has been added. Lines 306, 314, 318, 322, 325.

REVIEWER #2: 16) Tables 1-4 have very long descriptions (several times duplicated from the Materials and Method section), should be downsized, and placed as footnotes.

RESPONSE: The Table titles and footnotes have been edited as suggested.

REVIEWER #2: 17) Supplementary Information: Figure S2 correct to “Figure 5A, 5B and 5C.

RESPONSE: Corrected.

---

## [Editor Report · Decision Letter 1]

2 Jan 2024

Pharmacogenomics of poor drug metabolism in greyhounds: Canine P450 oxidoreductase genetic variation, breed heterogeneity, and functional characterization

PONE-D-23-24955R1

Dear Dr. Court,

We’re pleased to inform you that your manuscript has been judged scientifically suitable for publication and will be formally accepted for publication once it meets all outstanding technical requirements.

Kind regards,

Jed N. Lampe, Ph.D.

Academic Editor

PLOS ONE

---

## [Editor Report · Acceptance letter]

24 Jan 2024

PONE-D-23-24955R1 

PLOS ONE

Dear Dr. Court, 

I'm pleased to inform you that your manuscript has been deemed suitable for publication in PLOS ONE. Congratulations! Your manuscript is now being handed over to our production team.

Kind regards, 

on behalf of

Dr. Jed N. Lampe 

Academic Editor

PLOS ONE